# STABLE COGNITIVE MAPS FOR PATH INTEGRATION EMERGE FROM FUSING VISUAL AND PROPRIOCEPTIVE SENSORS

## ABSTRACT

Spatial navigation in biological agents relies on the interplay between allothetic (visual, olfactory, auditory, . . . ) and idiothetic (proprioception, linear and angular velocity, . . . ) signals. How to combine and exploit these two streams of information, which vastly differ in terms of availability and reliability, is a crucial issue. In the context of a new two–dimensional continuous environment we developed, we propose a direct-inverse model of environment dynamics to fuse image and action related signals, allowing reconstruction of the action relating the two successive images, as well as prediction of the new image from its current value and the action. The definition of those models naturally leads to the proposal of a minimalistic recurrent architecture, called Resetting Path Integrator (RPI), that can easily and reliably be trained to keep track of its position relative to its starting point during a sequence of movements. RPI updates its internal state using the (possibly noisy) self-motion signal, and occasionally resets it when the image signal is present. Notably, the internal state of this minimal model exhibits strong correlation with position in the environment due to the direct-inverse models, is stable across long trajectories through resetting, and allows for disambiguation of visually confusing positions in the environment through integration of past movement, making it a prime candidate for a **cognitive map**. Our architecture is compared to off-the-shelf LSTM networks on identical tasks, and consistently shows better performance while also offering more interpretable internal dynamics and higher-quality representations.

## 1 CONTEXT

**The Path Integration task.** Path Integration (PI), a task in which an agent has to integrate information about a sequence of movements to keep track of the distance between its current and initial positions, has been extensively studied both in rodents (Etienne & Jeffery, 2004; McNaughton et al., 2006), and in artificial systems (Arleo et al., 2000; Banino et al., 2018; Zhao et al., 2020), and is thought by many to be an essential ingredient in the elaboration of **cognitive maps** (Tolman, 1948; Redish & Touretzky, 1997), that is, internal representations of the spatial structure of an environment capable of supporting navigation tasks (Golledge, 2003). Path Integration is particularly relevant from the point of view of representation learning as it relies on the interplay between qualitatively different inputs, a subject known as multi-modal learning and recently reviewed by Summaira et al. (2021). Those inputs can be broadly categorized into two groups. On the one hand, **idiothetic** signals, such as velocity (Kropff et al., 2015), head direction (Taube et al., 1990), memory of past trajectories (Cooper et al., 2001) or reafferent copies of motor signals (Iacoboni et al., 2001), which are generated by the agent itself. On the other hand, **allothetic** cues, *e.g.* provided by vision (Etienne et al., 1996), olfaction or "whisking" in mice (Deschênes et al., 2012) are intrinsically related to the external environment.

The simplest solution to implement PI would be an integrator network that takes as inputs proprioception signals, or, equivalently, the agent's time-dependent velocity. While the theory of integrator networks and the representations that emerge have been well studied (Seung, 1996; Fanthomme & Monasson, 2021), such a solution suffers from two major limitations. First, the accumulation of errors across the trajectory, either coming from imperfect sensor information or from imperfect

integration, would make it unsuitable to represent Path Integration on arbitrarily long trajectories. Second, even if integration could be done without any error, representations constructed from proprioceptive information only would depend on the sequence of relative movements, and would be inadequate to the establishment of allocentric cognitive maps.

It is therefore crucial to understand how allothetic cues can be fused with self-motion information to achieve accurate PI, and to provide appropriate support for representations informative about the absolute position of the animal in its environment. This question has already been addressed in several works. Uria et al. (2020) introduced multiple recurrent neural networks (RNN) to build sophisticated 3D cognitive maps, with a variety of neurons displaying sensory-correlate features analogous to the ones encountered in cortical and hippocampal cell populations. Bicanski & Burgess (2018) proposed a model for the interactions between multiple brain areas concurring to the production of high-level spatial representations. In the field of robotics, Simultaneous Localization And Mapping systems based on Deep Learning are an active topic of research and show promising performance in key benchmarks of 3D navigation (Gupta et al., 2019; Zhang et al., 2020; Chaplot et al., 2020).

The objective of the present work is to address the issue of PI in the simplest possible setup, from the environment and network points of view, allowing for both good performance and interpretability. While our goal is not to provide a state-of-the-art method, *e.g.* directly applicable to robotics, we believe that such conceptual and (over)simplified approaches are valuable to help answer open questions about PI, such as its supervised or unsupervised nature, and its relevance to RL. In addition, the discrepancies between the representations built by our network and its natural biological counterparts may shed light on the additional functional and structural constraints acting on the latter.

**The environment, associated sensors, and PI loss.** In order to study PI in a simple and controlled setting, we developed a continuous 2–dimensional environment, which follows the basis of the OpenAI Gym specification (Brockman et al., 2016) to allow other researchers to reuse it in their experiments. This environment, detailed in Appendix A and Figure 1 includes a certain number of colored markers, which will act as landmarks to allow the agent to determine its absolute position. It also includes walls, which will impede some movements and restrict visibility. Movement and perception in this environment corresponds to a top-down perspective, similar to what could be found in a Pac-Man game, centered on the current position of the agent. This setup is limited compared to real three-dimensional environments, such as the ones based on Minecraft (Johnson et al., 2016) or Doom (Wydmuch et al., 2018). However, the resulting simulations are much faster and easier to run, and our framework is convenient for the study of sensor fusion. In addition, primates could be trained and monitored while performing a similar task, with eyes fixated on a screen displaying the environment and moving via a joystick; this would provide a direct comparison between artificial agents and biological ones and allow for a better understanding of both (Yang & Wang, 2020).

The two sensors that we want our agent to combine are: 1) a noisy copy of the action $\boldsymbol{a}^{re} \equiv \boldsymbol{a}^{tr} + \epsilon \, \mathbf{u}$, where $\mathbf{u}$ is a unit Gaussian vector, and $\epsilon$ the level of noise. This **reafferent action** represents the proprioceptive signal, in that it does not depend on the state of the environment; 2) a retinal signal $\boldsymbol{s}$, which represents the allocentric information, and depends on the position of the agent (Figure 1, Right).

This retinal state mimics, to some degree, the activity of a biological retina such as the one of our hypothetical monkey: an array[1] of Difference of Gaussians retinal cells is centered on the position of the agent; the activity of each cell is computed by summing over the currently visible landmarks the activity they each elicit, which depends on their distances to the center of the cell **receptive field**. More details on the retina, notably on the optimal linear decoding of position from the activity can be found in Appendix B. For this sensor, we consider two possible types of errors: (1) the retina receives no information, similar to what would happen if the screen flickers or the animal closes its eyes; (2) the retinal state is correct, but at some point along the visual processing pipeline the obtained representation is "shuffled" between neurons. This second type of errors is meant to represent a form of temporal multiplexing (Akam & Kullmann, 2014) in the corresponding population. Depending on time steps, cells participate in the visual processing pipeline, or in other cognitive tasks; in the latter case, we would expect the population activity to have similar distribution across neurons, but no correlation with the visual representation, which is here achieved through reshuffling.

---

[1] In practice, we use three superimposed arrays, one for each RGB color channel, see Appendix B.

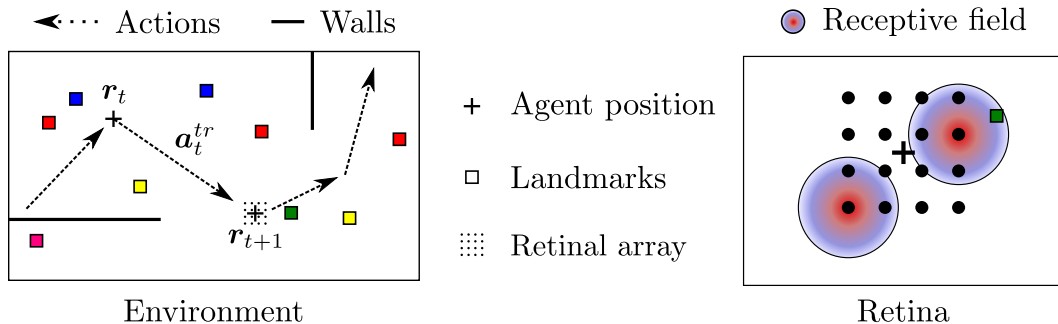

Figure 1: Presentation of our top-down perspective, two–dimensional continuous environment. Left: At each time step, the agent moves between positions $r_t$ and $r_{t+1}$ by performing an action $a_t^{tr}$. The image it perceives through its retina, now centered on the new position $r_{t+1}$, is modified accordingly, as the "landmarks" now occupy different positions with respect to the center of the retinal array. Right: Each neuron in the retinal array has an associated receptive field of the "Difference of Gaussians" type (for clarity, we represent only two); depending on the position of the landmark with respect to its receptive field, each neuron will be more or less activated, generating the "retinal state" that we will consider in the following as the "observation" received from the environment

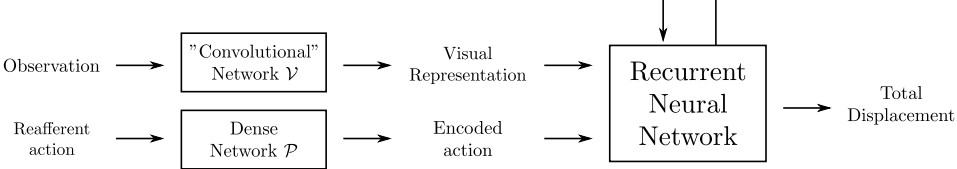

Figure 2: Shared structure of the models of Path Integration. The signals coming from the allocentric and proprioceptive sensors (respectively, the retinal activity and the reafferent action) are encoded through a first set of Neural Networks, before being used as inputs to a Recurrent Neural Network, whose output will be the predicted total displacement.

Based on these sensory signals, the agent has to estimate the displacements $\boldsymbol{\Delta r}_t$ from its starting point at all times $t$, see output of the PI network in Figure 2. We quantify the PI error along a trajectory of $T$ steps through the loss

$$\mathcal{L}^{PI} = \sum_{t=1}^{T} \left\langle \left( \boldsymbol{\Delta r}_t - \sum_{k=1}^{t} \boldsymbol{a}_k^{tr} \right)^2 \right\rangle, \qquad (1)$$

where $\langle \cdot \rangle$ represents the average over trajectories.

**Network architectures.** Since our PI task requires propagation of information from one time step to the next, it is not suitable for Multi-Layer Perceptron types of networks, which hold no memory of the previously received inputs, but can be handled with a Recurrent Neural Network (RNN). In the following we will consider two broad categories of RNNs, with variable architectures and training procedures: (1) off-the-shelf Long Short-Term Memory modules (Hochreiter & Schmidhuber, 1997); (2) a minimal RNN based on the idea of direct-inverse environment models defined in Section 2, which we call the Resetting Path Integrator (RPI) and introduce in Section 3. Precise architectures are presented in Appendix C, and a sketch of the common structure shared by all our networks is presented in Figure 2.

A natural approach to multimodal PI consists in simply concatenating non-recurrent encodings of action and visual inputs, and feeding the resulting joint representation to a Recurrent Neural Network trained on the PI loss (1). However, as reported in the following, these initial attempts yielded unsatisfying solutions that 1) failed to perform resetting (see Section 3) when an image was available, and 2) had internal states in the RNN that were correlated only with displacement from the start of the trajectory, and not with the absolute position in the environment. In order to foster the emergence of allocentric representations of the environment, we now introduce the concept of

direct–inverse models, and their associated losses. Direct–inverse models impose strong relationships between proprioceptive and visual signals, and as we shall see, lead to a natural approach to performing PI using those two qualitatively distinct streams of information.

## 2 DIRECT-INVERSE ENVIRONMENT MODELS

Evidence for **internal models** of environments has been found both in mammals (Ito, 2018) and in humans (Wolpert et al., 1998), notably within the Purkinje Cells of the Cerebellum, and have been hypothesized to be relevant for a wide range of motor (Wolpert & Miall, 1996; Kawato, 1999) and reasoning (Merfeld et al., 1999) behaviors. They have also been studied in the field of Reinforcement Learning, notably by Anderson et al. (2015), who showed that prediction of environment dynamics is an efficient pretraining step, by Pathak et al. (2017), who used the error in this model as a form of "curiosity" signal to encourage exploration, Corneil et al. (2018), who built a tabular model of an environment for use in an explicitly model-based planning algorithm, and Ha & Schmidhuber (2018), who used an internal model to allow the agent to learn from trajectories it "dreams" rather than from direct interaction with the environment, a formalism that could explain the observed coordinated replays of place and grid cells in rats (Ólafsdóttir et al., 2016).

These models can be formalized using the vocabulary of Partially Observable Markov Decision Processes (Sutton & Barto, 1998) used in Reinforcement Learning: the "hidden" state of the environment is the agent's absolute position; the observation is the retinal signal $s$ (see Section 1 for details), the "action" $a_t^{tr}$ at time $t$ is the displacement in the environment from time $t$ to $t+1$[2]. Models of the environment are defined on **transition tuples** $\tau = (s_t, a_t^{tr}, s_{t+1})$[3] and aim at predicting one of its components from the other two:

- the **direct** model $\mathcal{D}$ estimates the next state from the current one and the action:

$$\mathcal{D} : (s_t, a_t) \mapsto \overline{s_{t+1}}. \tag{2}$$

- the **inverse** model $\mathcal{I}$ estimates the action that relates two states:

$$\mathcal{I} : (s_t, s_{t+1}) \mapsto \overline{a_t}, \tag{3}$$

where $\langle \cdot \rangle$ represents the average over transition tuples. In practice, this approach would be highly inefficient and noise-sensitive in the case where the observed states are of high dimension but contain little relevant information (*e.g.* images). It is often preferable to construct these models on **representations**, obtained for example via a Convolutional Network $\mathcal{V}$; similarly, we introduce a Multi-Layer Perceptron (MLP) $\mathcal{P}$ that will map the two-dimensional action $a_t$ to a vector of the same dimension as the representation $\mathcal{V}(s_t)$; the resulting computation graph is presented in Figure 3, and the detailed architecture of the individual modules can be found in Appendix C.

To train these models we introduce two loss functions, computed from transition tuples:

$$\mathcal{L}^D(\mathcal{V}, \mathcal{P}, \mathcal{D}) = \left\langle \left[ \mathcal{V}(s_{t+1}) - \mathcal{D}(\mathcal{V}(s_t), \mathcal{P}(a_t)) \right]^2 \right\rangle, \tag{4}$$

$$\mathcal{L}^I(\mathcal{V}, \mathcal{I}) = \left\langle \left[ a_t - \mathcal{I}(\mathcal{V}(s_{t+1}), \mathcal{V}(s_t)) \right]^2 \right\rangle. \tag{5}$$

It should be noted that training the direct model $\mathcal{D}$ alone using the loss $\mathcal{L}^D$ of eqn (4) results in a trivial representation scheme in which all observations are mapped to the null vector $\mathbf{0}$. The inverse model $\mathcal{I}$ can be independently trained, but will generate irregular representations. When training $\mathcal{D}$ and $\mathcal{I}$ all together, the direct loss acts as a regularization, while the inverse loss breaks the symmetry required to converge to the trivial direct model. This yields a spatially structured representation of states, on which the direct operator acts non-trivially. More details on these representations can be found in Appendix D.

---

[2]The action could be considered a part of the observation, and the "partial observability" comes from the noise on these two as described in Section 1

[3]We do not include a reward signal as these experiments aim at mimicking "free foraging", in which the agent randomly explores an environment without explicit incentive to do so. The influence of a reward on representations will be the subject of a follow-up study.

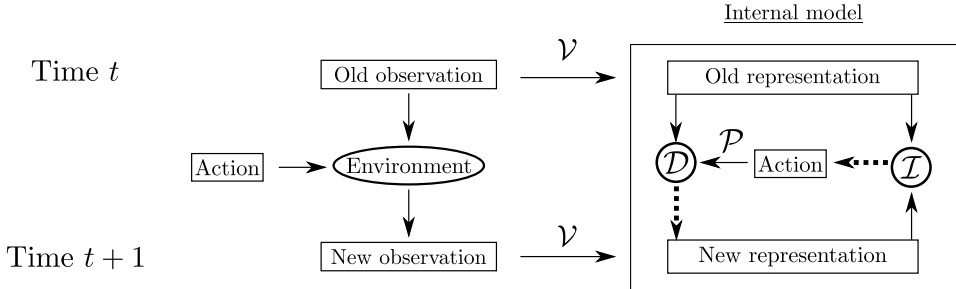

Figure 3: Overview of the direct-inverse model architecture, in which operators acting on internal representations aim at reproducing the dynamics of an environment. Dotted arrows indicate that the module they come from is trained to output the quantity they point towards (eqs. 4, 5).

## 3 RESETTING PATH INTEGRATOR FROM DIRECT-INVERSE MODELS

The direct and inverse models can straightforwardly be combined to create a RNN capable of Path Integration, which we will call Resetting Path Integrator (RPI) in the following, and which is summarized in Figure 4. The internal state $H$ is initialized with two concatenated copies of the initial observation [4]:

$$H_0 = \Big(\mathcal{V}(s_0) \, ; \, \mathcal{V}(s_0)\Big).$$ (6)

Then, at each time step, the internal state is updated using the direct model $\mathcal{D}$

$$H_{t+1} \equiv \Big(h_{t+1} \, ; \, \mathcal{V}(s_0)\Big) = \Big(\mathcal{D}(h_t, \mathcal{P}(a_t)) \, ; \, \mathcal{V}(s_0)\Big),$$ (7)

and the displacement $\Delta r_t$ is computed by applying the inverse model $\mathcal{I}$ between the updated and non-updated versions of the initial observation:

$$\Delta r_t = \mathcal{I}(h_t, \mathcal{V}(s_0)).$$ (8)

While this approach suffers *a priori* from the same accumulation of errors as the direct movement integration, it also allows for an additional **resetting** mechanism, which was hypothesized by Prescott (1996) as a sufficient mechanism for spatial navigation: at any time-step, the agent could use the current visual observation to "correct" its internal state by disregarding the result of the direct model $\mathcal{D}$. To allow for this, as well as partial resetting [5], we introduce a **gating network** $\mathcal{G}$ that maps the current visual observation $s_t$ [6] to a scalar between 0 (no resetting) and 1 (full resetting) that is used to interpolate between the proposed new state and the representation of the current observation, yielding the revised version of the update eqn (7):

$$H_{t+1} = \Big(\mathcal{G} \circ \mathcal{V}(s_t) \, \mathcal{D}(h_t, \mathcal{P}(a_t)) + [1 - \mathcal{G} \circ \mathcal{V}(s_t)] \, \mathcal{V}(s_{t+1}) \, ; \, \mathcal{V}(s_0)\Big).$$ (9)

Of course, if reliable images were always available, the optimal solution would correspond to $\mathcal{G} = 0$ at all times, that is, to resetting at every time step and never using the direct model. In standard situations, where reliable visual information may be lacking, the recurrent nature of the network will allow for correct performance in-between resetting steps by keeping the internal state close to what the agent would observe from the environment. We therefore expect the internal state of the network to strongly depend on the current value of the position, but not on the trajectory used to get there [7], hence being a valid candidate for a cognitive map. More subtly, if the visual information

---

[4] The second copy, which will not be modified by the network dynamics, is used as explicit "memory" of the starting point and is essential to observe resetting (see Section 4 and Appendix E for more details).

[5] Another possible approach could have been to enforce total resetting by using $\mathcal{G}$ as a probability to choose the visual state, and to train this gate with a Reinforce-like algorithm. This setting seems less biologically relevant than ours and we did not investigate it further.

[6] Formally, this network could also take the current state $H_t$ as input, but in practice this made the training more unstable without any noticeable improvement in performance.

[7] In time steps where strong resetting occurs ($\mathcal{G} \sim 0$), this statement is true since the state is exactly the representation of the current observation.

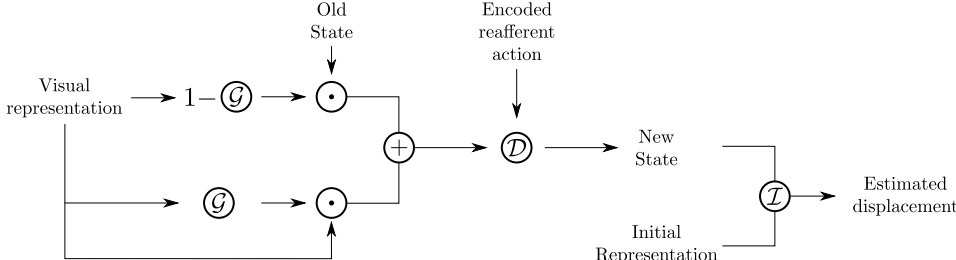

Figure 4: Minimal model for a Resetting Path Integrator, based on a Direct-Inverse model of environment dynamics. We assume that the agent is able to see correctly on the first step of the trajectory, and to keep a stable memory of this initial observation; this initial state is then updated by either using the direct model and the encoded reafferent action, or the new visual representation (resetting); the choice between those two behaviors is determined by the gating module $\mathcal{G}$.

received is ambiguous, *e.g.* the local set of landmarks seen by the retina is the same as in another part of the environment, see Section 4.2, we expect that the internal state should be able to lift the ambiguity in the observations through integration of previous motion, bridging the gap between regions of reliable visual information, a phenomenon which we actually observe in experiments.

To combine the direct-inverse and PI losses, training is done on their weighted sum

$$\mathcal{L}^{tot}(\mathcal{V},\mathcal{P},\mathcal{D},\mathcal{I},\mathcal{G}) = \alpha_{PI}\mathcal{L}^{PI}(\mathcal{V},\mathcal{P},\mathcal{D},\mathcal{I},\mathcal{G}) + \alpha_D\mathcal{L}^D(\mathcal{V},\mathcal{P},\mathcal{D}) + \alpha_I\mathcal{L}^I(\mathcal{V},\mathcal{I}) \qquad (10)$$

computed on random trajectories (see Appendix C for more details on the parameters).

## 4 RESULTS

In this section, we will analyze both the performance and the representations that emerge in networks trained on Path Integration loss of eqn. (1), using the environment layout represented in Figure 5, by looking at five different metrics. First, the average path integration error, computed (1) on short trajectories ($T = 5$), during which no image is presented to the network and (2) on long trajectories ($T = 100$) with images available every five steps. We expect short and long-term errors to be of the same order of magnitude in the case of a network that can perform resetting, while the latter will be much larger than the former if no resetting behavior has been learned. Then, the average (over neurons) of the *coefficient of determinations* $R_i^2$ for the linear regression of the absolute the activity of individual neurons $i$ participating to (3) the visual representation or (4) the internal state from the absolute position, and (5) of the (internal-state) neuron $i$ from the relative displacement within the trajectory. These individual $R_i^2$ scores are found to be close to 1, either for the absolute or the relative positions, indicating whether neuron $i$ is carrying allocentric or egocentric representation. Notice that $R^2 \sim 0$ would not mean that the neuron state would not convey any positional information, but that the latter would not be accessible to a linear decoder.

We will compare these five metrics under different training conditions: a "vanilla" LSTM model and our RPI model, trained on the PI loss in eqn. (1) only; an "improved" LSTM model[8] and our RPI model, trained end-to-end with the direct/inverse losses[9].

### 4.1 PERFORMANCE OF PATH INTEGRATORS AND NATURE OF REPRESENTATIONS

The results on the five metrics obtained in the snake-path environment of Figure 5 are reported in Table 1; a similar experiment carried out in a more complicated layout is presented in Appendix G. Four conclusions can be drawn from those results: (1) Both recurrent structures (LSTM and RPI) are able to learn resetting behaviors, yielding similar short and long-term errors, see Figures 5 and 6; (2) Training with Direct and Inverse losses (as regularization) is necessary for the emergence of resetting; (3) Our RPI model yields internal states with much higher positional tuning than LSTMs;

---

[8]Several variants of LSTM were considered with varying degrees of success, see Appendix E for details.

[9]Using these losses only for pretraining is possible, but leads to catastrophic forgetting, see Appendix F.

| | Resetting Path Integrator | | Long Short Term Memory | |
| --- | --- | --- | --- | --- |
| | All losses | No model losses | Vanilla | Improved |
| Error (short) | $0.021 \pm 0.017$ | $0.033 \pm 0.025$ | $0.014 \pm 0.011$ | $0.027 \pm 0.018$ |
| Error (long) | $0.026 \pm 0.022$ | $0.43 \pm 0.36$ | $0.16 \pm 0.15$ | $0.056 \pm 0.038$ |
| $R^2$ (visual) | $0.99 \pm 0.048$ | $0.11 \pm 0.091$ | $0.33 \pm 0.18$ | $0.96 \pm 0.085$ |
| $R^2$ (PI, absolute) | $0.98 \pm 0.053$ | $0.33 \pm 0.08$ | $0.3 \pm 0.097$ | $0.57 \pm 0.22$ |
| $R^2$ (PI, relative) | $0.35 \pm 0.077$ | $0.82 \pm 0.11$ | $0.77 \pm 0.21$ | $0.34 \pm 0.14$ |

Table 1: Comparison between our Resetting Path Integrator model and standard LSTM in the SnakePath environment. When trained without the model losses, both architectures fail to establish a proper resetting strategy, leading to higher error rates when tested on long trajectories ($T = 100$) than on short ones ($T = 5$), and the internal state during PI is a linear function of displacement along the trajectory, rather than of absolute position in the environment as is the case when model losses are used. Errors bars were estimated from 20 realizations of the training which differ both by initialization of the network weights, and drawn training trajectories.

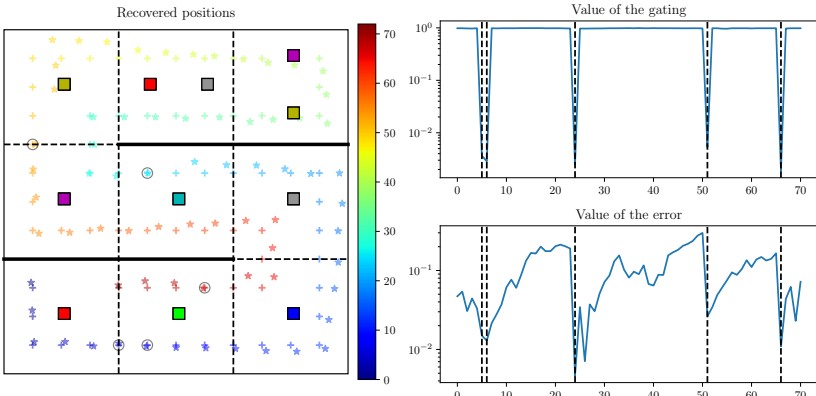

Figure 5: Example of Path Integration trajectory Left: Crosses represent the true position of the agent, while stars represent the one evaluated through Path Integration; black circles are placed around the positions at which an image was provided; time along the trajectory is represented by the color of the symbols. Top right: logarithm of the value of the resetting gate as a function of time along the trajectory. Bottom right: error between the true and reconstructed position. Vertical dashed lines indicate the time-steps at which the image was available to the network and not corrupted. In this example, actions are not drawn from the "free foraging" random policy but chosen to force exploration of the entire environment to better evaluate generalization at long distances, and reafferent actions are exact, so that errors are due only to the network itself.

(4) Representations depend on absolute position in networks that perform resetting, and on relative position along the trajectory in networks that do not (see Appendix H for details).

Despite RPIs being less expressive than their LSTM counterparts, they do not seem to achieve significantly worse performance (although more hyperparameter optimization would be required to confirm this statement). In addition, RPIs converge to solutions that are more easily interpretable, both in terms of tuning to the absolute position (illustrated by the higher $R^2$ scores), and of gating dynamics, see Figure 5. In Appendix I, we show that the value of the gate in a trained RPI is directly related to the cognitive mechanism of resetting, with the strength of resetting increasing when the training conditions incorporate more noise in the proprioceptive signals, as well as when the visual representations are more and more perturbed. In LSTMs, however, the reset and input gates are only weakly correlated with resetting, and instead might contribute to the computation of the direct model, see Appendix J for details.

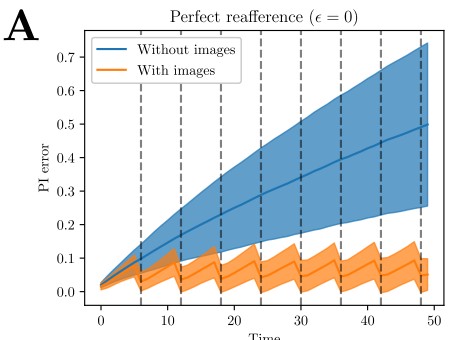 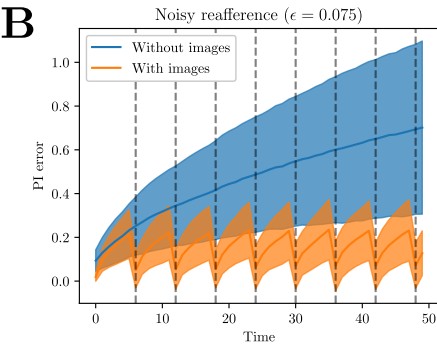

Figure 6: Path Integration errors achieved by our Resetting Path Integrator, with occasionally available retinal images (orange) and without images (blue). **A**: the reafferent action (proprioceptive signal) is exactly equal to the true one. **B**: a small amplitude Gaussian noise $\epsilon$ is added to the reafferent action. Dashed vertical lines indicate steps at which images were presented, kept equal across 512 trajectories for each of the 8 networks used in the averaging. The qualitative agreement between those two plots, as well as results from Appendix I, suggest that our procedure is robust to small reafference errors, which have the same effect as direct model errors.

## 4.2 DISAMBIGUATION OF AMBIGUOUS ENVIRONMENT BY RPI REPRESENTATIONS

Next, we considered a highly ambiguous situation where two rooms, located at the opposite ends of the environment, are designed to provide strictly identical visual cues. In that case, inverse models of the full environment give very large reconstruction errors when either of the images are located within one of these rooms.[10]

However, training a Resetting Path Integrator on this environment in an end-to-end fashion remains possible, as shown in Appendix K. The resulting networks exhibit three important properties: 1) the internal states observed during Path Integration are different in the two ambiguous rooms, as illustrated in Figure 7; 2) the PI error does not show any noticeable increase when the agent enters one of the ambiguous rooms, see Supplementary Figure 18; 3) the resetting mechanism is not triggered for images coming from the ambiguous rooms, as illustrated in Supplementary Figure 19.

The last observation was to be expected: as visual representations are identical between the two rooms, performing a resetting would, on average, result in a loss of spatial information with respect to keeping the state updated through the direct model. The first and second observations are nontrivial. To correctly perform Path Integration in the ambiguous rooms, our RPI network created new states, differing from those coming from the visual cues, that aim at bridging the gaps between "visually informative" regions. These networks have therefore managed to construct a representation of absolute position in the environment that does not rely only on local landmarks, but also draws from proprioceptive information and effectively fuses these sensors; intuitively, the Path Integrator is able to differentiate between two visually identical rooms by remembering how it got there, a highly desirable property for cognitive maps.

## 5 CONCLUSION

**Results.**   In this study, we have demonstrated how a Recurrent Neural Network can be used to construct a cognitive map of a continuous spatial environment, by fusing unreliable proprioceptive and intermittently available external inputs through the task of Path Integration. We examined several ways of performing this fusion, using either off-the-shelf LSTM networks, or our proposed Resetting Path Integrator model, based on direct-inverse models of the environment dynamics, and including a single, scalar gating mechanism allowing for resetting (clearing) the internal state of the network and replacing it with an external signal. While all studied architectures and training procedures manage

---

[10]If we train and test only on images coming from adjacent rooms, an inverse model can perform well as there is no couple of images that would correspond to two different reconstructed positions. The resulting direct–inverse models are however not suitable for Path Integration without retraining, as their long-range performance is severely limited by the ambiguous rooms.

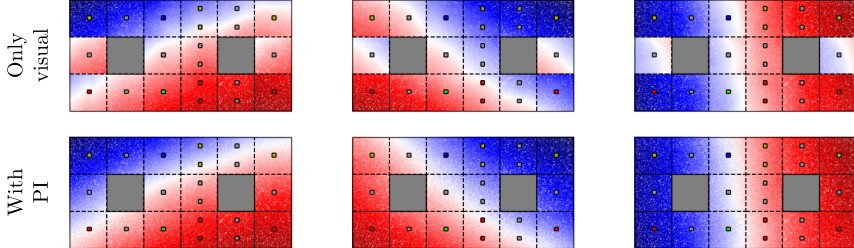

Figure 7: Comparison between representative neurons in the visual module $\mathcal{V}$ (top row) and the internal state $h$ observed during Path Integration (bottom row) as a function of position within our ambiguous environment. The "dynamic" representation constructed during PI lifts the ambiguity between the two opposite rooms of the middle row, which contain the same landmark and are surrounded by identical rooms. Each column represents the normalized activation of a single neuron.

to perform Path Integration on short trajectories, only those regularized through the addition of the direct-inverse losses learned to efficiently use resetting and achieved similar error levels on very long and on short trajectories, thus overcoming error accumulation. The idea of incorporating high-level knowledge about desirable aspects of the internal states dynamics through regularization is close to the one of Haviv et al. (2019). We observe that the internal neural states are qualitatively different between path-integrator networks that learn resetting and those that do not: while the former are very close to linear functions of the absolute position in the environment, the latter are closer to linear functions of the displacement along the trajectory, see Table 1. This subtle difference is crucial, and implies that only networks capable of resetting have learned a "cognitive map" (Tolman, 1948; Spiers & Barry, 2015), which could *a priori* be transferred towards other spatially structured tasks.

**Future directions.** Contrary to previous works on PI in artificial agents that used highly spatially-structured inputs, relying on hypothesis about the existence of either place cells (for example in Arleo et al. (2000); Banino et al. (2018)) or grid cells (Zhao et al., 2020), our approach does not make such assumptions, and is, to our knowledge, the first one to allow for study of the emergence of these representations during training. Our simplistic environment setup did not result in emergence of either of those types of cells, but instead on a "top-down" map, which accurately depicts the Euclidean (by opposition to topological) structure of the environment. In other words, positions that are close in the layout (viewed from above) are close also in the map, though they could be far from each other in terms of "minimum number of steps", *e.g.* when separated by a wall that would need to be walked around. It is then a logical next step to move towards more realistic environments, using real first-person view, and allowing for rotations and translations, *e.g.* Malmo (Johnson et al., 2016) or VizDoom (Wydmuch et al., 2018), whose interplay might be responsible for the particular coding scheme of place and grid cells (Harsh et al., 2020; Benna & Fusi, 2020).

Another major direction of research concerns the role of Path Integration as an end-goal: while it is assumed that such a task can be used to generate high-quality cognitive maps (as confirmed by our study), there is no evidence that this task is ever performed "intentionally". Preliminary experiments show that the recurrent representations constructed by the networks can be used for Reinforcement Learning tasks, such as goal-oriented navigation (moving towards a specific position in the environment), much more efficiently than through direct training. This result was to be expected, since representation learning is known to be a limiting factor in RL (Anderson et al., 2015). In a follow-up study, we will focus on incorporating the Path Integration loss (as well as the direct and inverse losses) as regularization terms in the policy learning algorithm; this approach is similar to the Intrinsic Curiosity Module of Pathak et al. (2017), in which the model errors were used as an exploration incentive, as well as zero-shot learning through environment models (Ha & Schmidhuber, 2018). All those methods can *a priori* be used at the same time.

Combining those two directions of research, making both tasks and environments more realistic, in particular close to what can be studied in live animals, will be key to understanding the intricacies of cognitive maps and of their relation to behavior.

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

## A  THE CONTINUOUS GRIDWORLD ENVIRONMENT

All experiments presented in this article were performed using the GridWorld environment class, defined in the environment.py file. While this environment is neither particularly efficient to run, as it is coded in pure python, nor very expressive, it still allows for a wide variety of interesting situations and is performant enough to not be an unreasonable bottleneck in most situations. It also follows the basic specifications of the OpenAi Gym framework, which makes it easy to extend it and test basic Reinforcement Learning tasks such as goal-driven navigation.

While we do not provide level editors or tools to procedurally generate new layouts, they can be added by hand in the environment register, under a new key corresponding to the map name, as a dictionary containing the following attributes:

- the number of rooms in the environment;
- a list containing the position of the room centers in the surrounding environment (the environment can be rescaled as a whole when instantiated, so it is easier to assume all rooms to be of size $1 \times 1$);[11]
- a list of room exits, such that the $i$-th component is a list containing all exits from room $i$. An exit is defined as a room edge (either vertical or horizontal) that is connected to another room edge, and the link between coordinates in the two rooms is established by giving the coordinates of the same physical point in the two rooms (the center of the common room edge);
- a list of possible item layouts, detailing what items are visible (and at which position from the room center) within each room, allowing us to simulate visual occlusion, or even special effects (such as switching the light off when the agent is in a particular room). All items are point-like light emitters, and only differ through their color.

Given the connection graph of the rooms and the item layout, the GridWorld class can be used to generate trajectories from sequences of actions, as well as to generate a human-interpretable rendering of the arrangement of rooms and items which can be used as a basis for more involved plots (most notably, trajectories and value of neuron activity as a function of position in the environment).

The inputs to the network, which we refer to simply as images, are obtained from a list of rooms and positions within these rooms, by using a retina of the type described in Appendix B.

We provide several environments and layouts, some of which were used only for preliminary tests but which we retain for the sake of completeness.

## B  RETINA

**Individual retinal fields**  We begin by introducing the Difference of Gaussians retinal neurons (Dayan & Abbott, 2001): their receptive fields have a **center** $c$, two widths $(\sigma_+, \sigma_-)$, and their activity $g$ when a single visual cue is present at distance $\delta r$ from the center is a difference of Gaussians:

$$g(\boldsymbol{r} = \boldsymbol{c} + \delta\boldsymbol{r}) = \frac{A}{\sqrt{2\pi}\sigma_+} \exp^{-\frac{\delta r^2}{2\sigma_+^2}} - \frac{B}{\sqrt{2\pi}\sigma_-} \exp^{-\frac{\delta r^2}{2\sigma_-^2}} \tag{11}$$

When two or more visual cues are presented in the image, the activation of the neurons will be the sum of the activations for each individual object.

**Retinal array**  We will consider as **retina** a square array of these retinal fields (see Figure 8), all with $A = B$ for simplicity. The value of $A$ is determined so that, on average over the position of a

---

[11]While this environment is designed to be used for 2–dimensional spatial navigation, we consider the room centers to have three components; the first two are the $(x, y)$ coordinates which can be modified by our two-dimensional actions, while the third one can be used to lift ambiguity between two rooms that would be located at the same position but would differ in another way meaningful for the environment (for example, to change the environment after the agent visits a particular room). This functionality is not used in the experiments presented in this study.

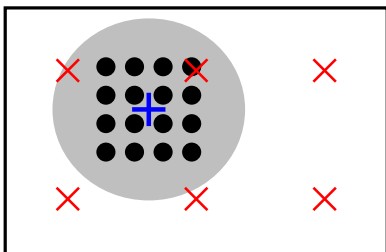

Figure 8: Example of retina: a regular square lattice of Difference of Gaussians fields. We describe as the "area of effect" of the retina the zone in which an object would contributes to the activity vector of the retina as a whole above some arbitrary threshold, *e.g.* $10^{-2}$.

single object within a square room of width 1, the average norm of the retina activity vector is equal to 1. For experiments, we will use an array of $64^2 = 4096$ cells; for the values of the two widths, we choose $(\sigma_+ = 0.4, \sigma_- = 0.5)$.

**Color retina**   In order to be able to differentiate between two objects, we associate to each a "color", i.e. a vector in $[0, 1]^3$. This color is perceived by the retina in the following way: there are three "copies" of our retina, one for each color "channel"; the object activates each "channel" proportionately to the object's value in that color. This makes it so that each position $r$ in the environments corresponds to one image $i$ of size $(3, 64, 64)$ for the $64 \times 64$–retinas that we use.

**Optimal linear reconstruction of an arbitrary function of position**   Let us consider the family of functions $\{(g_1(r), g_2(r), \ldots g_n(r)\}$, where $g_i(r)$ describes the activation of neuron $i$ when a cue is located at position $r$. A linear model of an arbitrary function $f$ (of the cue position) from the state of our retina can be written as a linear combination of those functions:

$$\hat{f}(r) = \sum_i \alpha_{f,i} g_i(r) \tag{12}$$

The associated reconstruction error on a domain $\mathcal{D} \subset \mathbb{R}^2$ can then be written:

$$\begin{aligned}
\mathcal{E}(\alpha_f) &= \int_{\mathcal{D}} \left[ \sum_i \alpha_{f,i} g_i(r) - f(r) \right]^2 \\
&= \int_{\mathcal{D}} \left[ \sum_{i,j} \alpha_{f,i} \alpha_{f,j} g_i(r) g_j(r) - 2 f(r) \sum_i \alpha_{f,i} g_i(r) + f(r)^2 \right] \\
&= \sum_{i,j} \alpha_{f,i} \alpha_{f,j} \int_{\mathcal{D}} g_i(r) g_j(r) - 2 \sum_i \alpha_{f,i} \int_{\mathcal{D}} f(r) g_i(r) + \int_{\mathcal{D}} f(r)^2 \\
&:= \sum_{i,j} \alpha_{f,i} \alpha_{f,j} I_{ij} - 2 \sum_i \alpha_{f,i} h_{f,i} + C
\end{aligned} \tag{13}$$

The minimum of this loss with respect to the parameters $\alpha_f$ satisfy:

$$\forall i, \frac{\partial \mathcal{E}}{\partial \alpha_{f,i}} = 2 \left[ \sum_j I_{ij} \alpha_{f,j} - h_{f,i} \right] = 0 \tag{14}$$

and therefore the optimal linear decoder is obtained as $\alpha_f = I_f^{-1} h_f$.

Numerical approximation of the integrals over $\mathcal{D}$ are easily obtained by averaging over measures on a grid lattice, up to some numerical precision limitations. This decoding is not particularly fast since determining $I$ is computationally expensive.

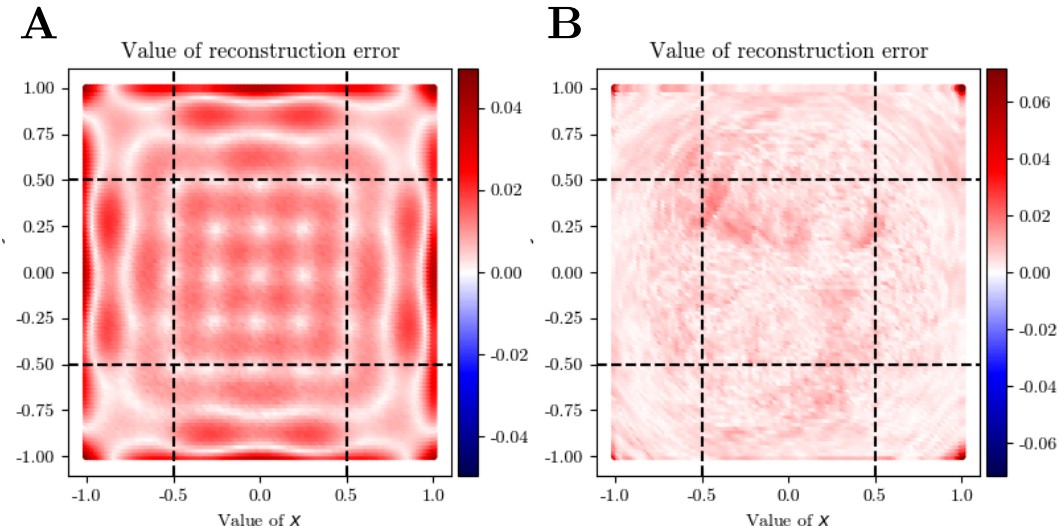

Figure 9: Reconstruction error on a grid with 100 subdivisions on $x$ and $y$ as a function of position, represented as a heatmap. On the left, the reconstruction error of the optimal linear decoder shows clear geometric patterns related to the geometry of the underlying array of cells. On the right, the reconstruction error for a "deep" 3-layer ReLU network. While the highest observed error is higher for this deep network, the error on all points except the corners is much lower than for the linear network. Additionally, the geometric patterns are not observed in that case.

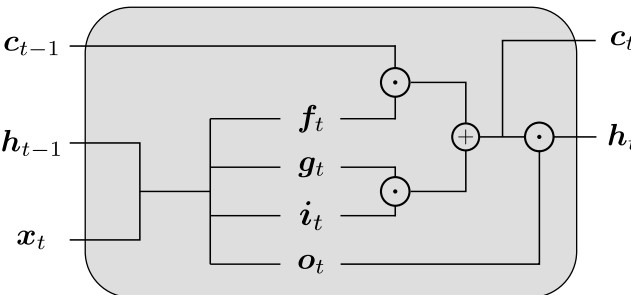

Figure 10: Computation diagram for a LSTM cell, as implemented in Paszke et al. (2019).

**Expected optimal performance** When the environment consists in a single room, containing a single object, we can easily reconstruct the relative position of the object with respect to the center of the retina using either the direct linear solving of the previous paragraph or gradient descent on a more parametrized structure, *e.g.* a dense or convolutional neural network. This experiment gives us an idea of the maximum performance that can be expected from any inverse model based on our retina.

When using a $64^2$ array spanning $[-0.5, 0.5]$ and reconstructing the position of an object placed arbitrarily in $[-1, 1]$, the Root Mean Square error for the optimal linear reconstruction is around $10^{-2}$, with clear geometric patterns in the errors, while the 3-Layers ReLU network achieves a performance closer to $10^{-3}$ by seemingly "smoothing" the aforementioned error patterns. These results are presented in Figure 9.

## C  NETWORK ARCHITECTURES AND TRAINING HYPERPARAMETERS

### C.1  ARCHITECTURES

The exact architectures used in our implementations are as follows:

- the convolutional networks $\mathcal{V}$ used to obtain the visual representations are:

  1. Input image with 3 channels, size $64 \times 64$ (from the retina)
  2. Convolution with 16 filters, kernel size of 5, stride of 3, padding of 2
  3. Convolution with 32 filters, kernel size of 5, stride of 5, padding of 2
  4. Flatten layer
  5. Dense layer with 512 outputs
  6. Dense layer with 512 outputs

- the action encoding networks $\mathcal{P}$ are:

  1. Input layer of size 2
  2. ReLU layer with 256 outputs
  3. Linear layer with 512 outputs

- the direct model networks $\mathcal{D}$ are:

  1. Input layer of size 1024 (concatenation of representation and action encoding)
  2. ReLU layer with 512 outputs
  3. Linear layer with 512 outputs

- the inverse model networks $\mathcal{D}$ are:

  1. Input layer of size 1024 (concatenation of two representations)
  2. ReLU layer with 256 outputs
  3. Linear layer with 2 outputs

- the gating networks $\mathcal{G}$ are:

  1. Input layer of size 512 (visual representation)
  2. ReLU layer with 128 outputs
  3. ReLU layer with 64 outputs
  4. Sigmoid layer with 1 output

It should be noted that our Recurrent Path Integrator models have number of parameters of the same order of magnitude as the off-the-shelf implementations of LSTM that we used (see Figure 10).

## C.2 TRAINING

In all experiments we present, the PI losses are computed on batches of 32 trajectories of length 40, with actions drawn from a two-dimensional uncorrelated Gaussian of standard deviation 1/2 and starting point chosen randomly at any position in any non-ambiguous room. The direct and inverse losses are computed on batches of 512 transitions, for which starting points and actions are chosen in the same way as for PI.

The relative weights in the total loss are 10 times higher for the direct inverse losses than for the PI loss as the former are smaller (these hyperparameters have not been optimized)

Training consists in 4000 steps of computing the losses, and performing one step of Adam optimization with uniform learning rates of $10^{-3}$ except for the forward model which is at $10^{-4}$ (no hyperparameter optimization was lead on these either). In most cases, losses only evolve marginally after the first hundreds of epochs.

We emphasize that since the environment, starting positions and actions are both continuous and random, no trajectory is ever seen twice by the network, so overfitting is not a concern (rather, we hope that the network meaningfull interpolates between what it has previously seen). However, we train the networks on a single environment, and the question of the capacity–resolution tradeoff (how the precision of PI is modified when several environments are learned at the same time) remains unaddressed in the present study and a meaningful future direction.

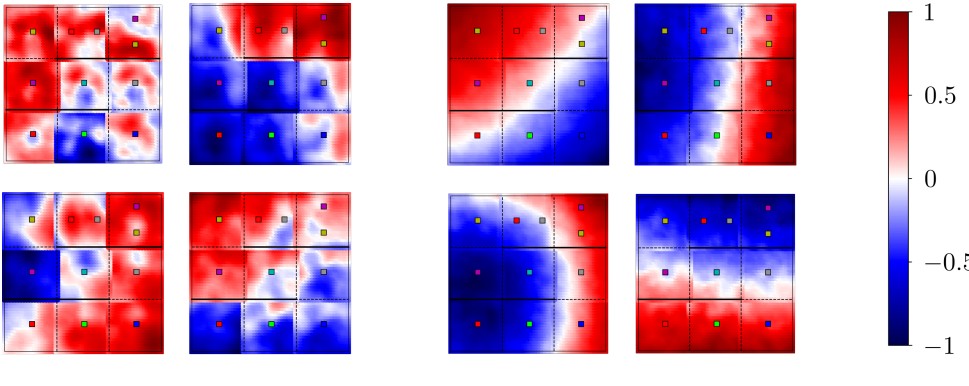

(a) Without forward model  (b) With forward model

Figure 11: Comparison between representations obtained after training the Direct-Inverse Model module in our environment. Each panel represents the normalized activation of a single neuron in the visual representation $\mathcal{V}(\boldsymbol{s})$, obtained at the end of the visual processing module, represented as a function of position within the environment through a color code presented on the right scale. The activities are of the same order of magnitude in all cases. When optimizing only the inverse loss (eq. 5), see panel (a), the representations can be spatially irregular; the introduction of the direct loss (eq. 4) smooths out the activities, see panel (b). This effect is quantified in Table 2.

Table 2: Comparison of the inverse model performance and the representation regularity between models trained with or without the direct loss. The addition of the direct loss shifts the distribution of $R^2$ scores between neuron activities and spatial position towards one, meaning it made some neuron activities closer to linear functions of position, which we argue is a desirable property in order to obtain transferable representations. While this shift is noticeable, it does not come with any appreciable change in inverse model performance. Means and deviations computed across 8 realizations.

|  | Error (training) | Error (generalization) | $R^2$ (visual) |
|---|---|---|---|
| Without direct | $0.017 \pm 0.01$ | $1.6 \pm 0.77$ | $0.95 \pm 0.089$ |
| With direct | $0.015 \pm 0.0091$ | $1.6 \pm 0.75$ | $0.83 \pm 0.19$ |

## D  INTERACTIONS BETWEEN DIRECT AND INVERSE LOSSES

As mentioned in the main text, training the inverse model without the direct one is possible, but the other way around is not as training in that case converges to a trivial solution where the representation module $\mathcal{V}$ and the direct model $\mathcal{D}$ always output $\boldsymbol{0}$. When training with both losses, we observe a slight but noticeable smoothing of the representations, as illustrated in Figure 11 and Table 2. This improvement in regularity does not however translate into any measurable difference in performance of the inverse model: the representations are made simpler by the direct loss (as expected when adding a regularization term), but they do not carry any more positional information. It should be noted that while "generalization" errors (computed on any couple of images, even if they can not be reached in a single transition and hence were never part of a transition tuple used in training) are large, the difference in position predicted remains qualitatively relevant (just with a lower precision).

## E  LSTM VARIANTS

In this section, we present our attempts at improving the performance of the "vanilla" LSTM architecture. Beyond basic hyperparameter tuning (no extensive optimization has been led due to prohibitive computational costs), we mostly considered modifications on initialization, and architecture:

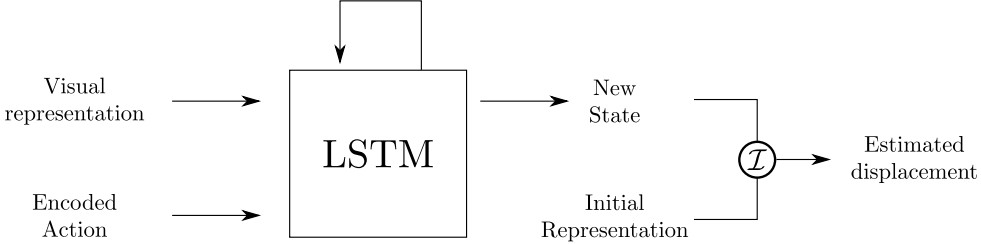

Figure 12: Computation diagram for the hybrid path integrator structure.

- the standard architecture corresponds to simply using a LSTM (see Figure 10) as the RNN in the computational graph of Figure 2. We considered two training schemes with this architecture:
  - the "vanilla" scheme trains this network from scratch on the PI loss. It yields good integration properties, but fails to learn resetting behaviors (results presented in main paper).
  - the "pretrained" scheme initializes the "encoders" (both for the image and the action) using the ones of a Resetting Path Integrator trained with all losses (since they exhibit the cleanest representations). The results are very similar to the "vanilla" scheme.
- the "hybrid" solutions have the computation diagram of Figure 12, which corresponds to using the LSTM only to update the internal state, replacing the combination of the direct model $\mathcal{D}$ and gating module $\mathcal{G}$. This architecture has the advantage of explicitly using the initial representation as a form of "anchor", which seems in practice to help training converge to resetting behaviors. We considered several training schemes using this architecture:
  - the "default" scheme trains this network from scratch on the PI loss, yields similar result to vanilla LSTM.
  - the "pretrained" initializes the "encoders" (both for the image and the action) using the ones of a Resetting Path Integrator trained with all losses; it reliably achieves resetting, but also has lower precision on short trajectories.
  - the "scratch" scheme does not do any initialization, but adds a "direct" module (same architecture as the ones of our Resetting Path Integrator), defines the direct and inverse losses using it and the $\mathcal{I}$ module that outputs the displacement, and uses those losses as regularization. This scheme often manages to find resetting solutions, but requires more care in hyperparameter tuning to converge properly to a resetting solution.
  - the "improved" scheme is similar to scratch, but also initializes the encoders. This scheme is included in the main text, and it reliably achieves resetting.

In Table 3, we present results for the aforementioned architecture that were not included in the main text, Table 1. It should be noted that even schemes that yield solutions that exhibit resetting do not necessarily have higher levels of positional tuning, which we argue still makes them less convincing candidates for transferable cognitive maps than our Resetting Path Integrator model.

## F    CURRICULUM LEARNING AND CATASTROPHIC FORGETTING

In this section, we consider the case in which the direct-inverse model of the environment is trained first, using transition tuples, without any consideration of Path Integration.

After this initial pretraining, we introduce those weights into the full Resetting Path Integrator network, and consider three different ways of training the PI task:

- **A**: we optimize only the weights of the resetting gate $\mathcal{G}$, and use only the Path Integration Loss.
- **B**: we optimize all weights in the network, including those that were initialized from the pretraining, still using only the Path Integration Loss.

Table 3: Comparison between the different LSTM variants we considered on the SnakePath environment, in terms of both errors and representation correlation with position, see main text for details. Means and errors computed on 8 realizations.

| | Standard | Hybrid | | |
| --- | --- | --- | --- | --- |
| | Pretrained | Default | Pretrained | Scratch |
| Error (short) | $0.01 \pm 0.0064$ | $0.013 \pm 0.0086$ | $0.034 \pm 0.02$ | $0.022 \pm 0.014$ |
| Error (long) | $0.091 \pm 0.088$ | $0.11 \pm 0.099$ | $0.043 \pm 0.026$ | $0.042 \pm 0.03$ |
| $R^2$ (visual) | $0.46 \pm 0.2$ | $0.63 \pm 0.24$ | $0.91 \pm 0.14$ | $0.94 \pm 0.11$ |
| $R^2$ (PI, absolute) | $0.27 \pm 0.11$ | $0.29 \pm 0.11$ | $0.63 \pm 0.19$ | $0.58 \pm 0.21$ |
| $R^2$ (PI, relative) | $0.69 \pm 0.26$ | $0.72 \pm 0.24$ | $0.24 \pm 0.092$ | $0.34 \pm 0.14$ |

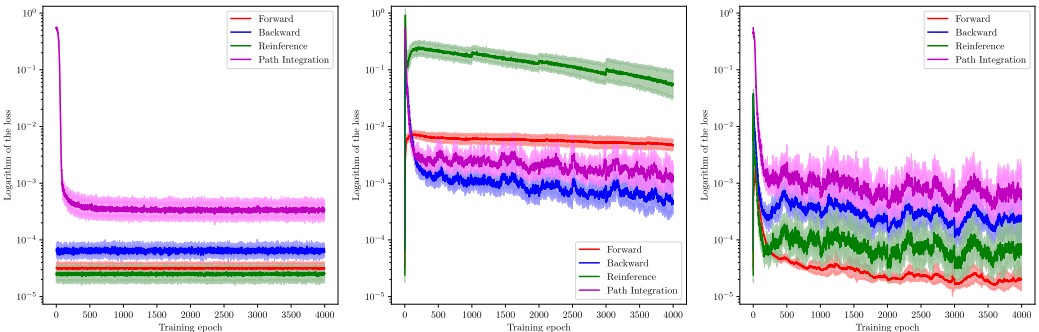

Figure 13: Evolution of the different losses when training a Resetting Path Integrator, whose visual, direct and inverse modules were pretrained using transition tuples from the environment, in the three different protocols described in the text.

- **C**: we optimize all weights in the network, but do Gradient Descent on a sum of all losses (Path Integration, direct and inverse), as we would in end-to-end training.

The results are presented in Figure 13, and show that while the final level of Path Integration error is close between the different protocols, retraining all parameters of the model on the Path Integration loss only (protocol **B**) produces noticeable deterioration in the quality of the Direct-Inverse model, an example of the catastrophic forgetting phenomenon (Kirkpatrick et al., 2017). Independently of the choice of loss on which Gradient Descent is performed, optimizing on the parameters of the Direct-Inverse model produces much more chaotic evolution of the loss.

## G    ERRORS AND SPATIAL CORRELATIONS ON THE DOUBLEDONUT ENVIRONMENT

We present in Table 1 the values of errors and spatial correlations measured in networks trained on a second environment layout, slightly different from the one used in the main text, and which we call DoubleDonut. This environment has the general structure of its ambiguous variant (represented in Figure 18), except that the objects in the left- and right-most rooms of the middle row are different in the case of the non-ambiguous version that we consider here. The conclusions of this study are identical to the ones presented in the main text.

## H    REPRESENTATIONS IN ABSOLUTE AND RELATIVE COORDINATES

In order to complement the $R^2$ values presented in main text Table 1, we report the value of neuron activations (in the internal state, observed during Path Integration) as a function of absolute position in the environment (Figure 14) and as a function of position within the trajectory (Figure 15)

Table 4: Comparison between our Resetting Path Integrator model and standard LSTM in the DoubleDonut environment. As was the case in the SnakePath environment, models trained without the direct-inverse losses fail to learn how to perform resetting and show lower levels of spatial structure in their representations.

| | Resetting Path Integrator | | Long Short Term Memory | |
| --- | --- | --- | --- | --- |
| | All losses | No model losses | Vanilla | Improved |
| Error (short) | $0.026 \pm 0.019$ | $0.035 \pm 0.026$ | $0.015 \pm 0.0086$ | $0.032 \pm 0.019$ |
| Error (long) | $0.032 \pm 0.023$ | $0.46 \pm 0.41$ | $0.15 \pm 0.14$ | $0.051 \pm 0.033$ |
| $R^2$ (visual) | $0.97 \pm 0.068$ | $0.16 \pm 0.12$ | $0.36 \pm 0.16$ | $0.91 \pm 0.13$ |
| $R^2$ (PI, absolute) | $0.96 \pm 0.073$ | $0.29 \pm 0.063$ | $0.27 \pm 0.086$ | $0.68 \pm 0.19$ |
| $R^2$ (PI, relative) | $0.34 \pm 0.058$ | $0.85 \pm 0.14$ | $0.74 \pm 0.22$ | $0.26 \pm 0.083$ |

All losses          PI loss only

Figure 14: Activity of four representative neurons in the internal state population of an RPI trained with (left) or without (right) the model losses, as a function of absolute position in the environment. Only the ones trained with those losses (and performing resetting) are close to a linear function of absolute position.

for our Resetting Path Integrator model, trained with or without the direct–inverse losses (and consequently, respectively displaying resetting or not). We find that non-resetting representations are linear functions of displacement, as expected from an integrator network (see Fanthomme & Monasson (2021)), while resetting representations are linear functions of absolute position, which makes them much more relevant as cognitive maps. Interestingly, we note that (for the resetting network on the left of Figure 15), points that correspond to "extreme" displacements seem more correlated with trajectory coordinate than points with small trajectory coordinates; this is to be expected since extreme displacement points necessarily lie on the edge of our environment (to get a displacement of $-3$ in the $x$ direction, the agent necessarily started on the right side of the environment and finished on the left side), so that for those points trajectory coordinates and absolute coordinates are correlated.

## I  GATING STRENGTH IN A RESETTING PATH INTEGRATOR

While all training conditions we investigated lead to resetting behaviors, the mean value of the gating obtained from images of the environment was observed to vary drastically, from $10^{-1}$ to $10^{-3}$, while the level of Path Integration errors remained mostly unchanged. Our understanding for this phenomenon is the following: the minimum level of achievable error is the same for all training conditions, and related to the limitations of the retinal array detailed in Appendix B; therefore, the

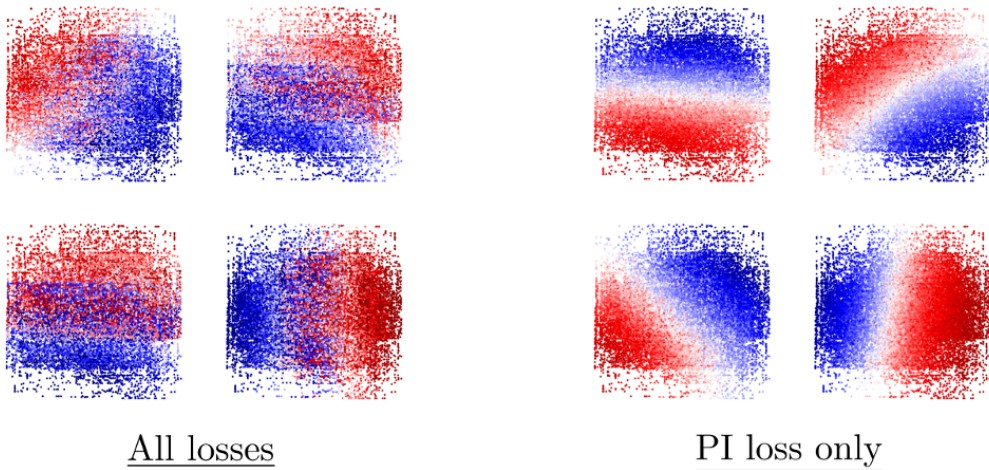

Figure 15: Activity of four representative neurons in the internal state population of an RPI trained with (left) or without (right) the model losses, as a function of position along the trajectories. Only the ones trained without those losses (and not performing resetting) are close to a linear function of position within the trajectory.

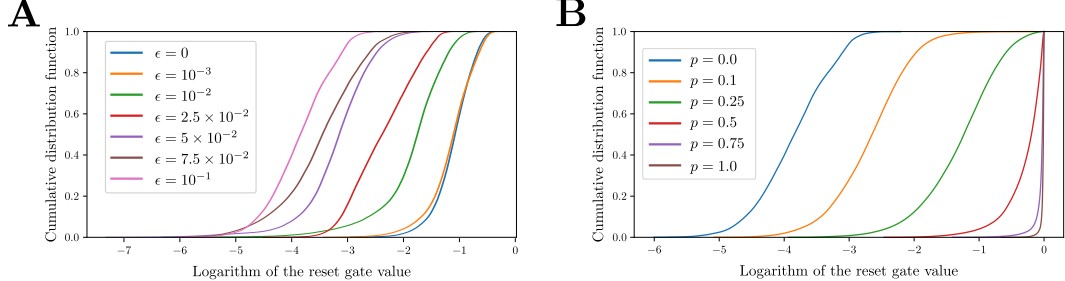

Figure 16: Cumulative distribution function of the natural logarithm of the reset gate $g$ across the environment in different conditions. **A**: Varying the level of noise $\epsilon$ in the reafferent action during training. As expected, high levels of noise favor strong resettings, hence lower values of $g$. **B**: Varying the level of perturbation $p$, defined as the fraction of neurons in the representation that were randomly reshuffled, at test time. As representations are increasingly perturbed, they become less similar to ones that come from the environment, and we expect the network to reset its state less strongly as the expected benefit from such a resetting decreases. Both panels present results aggregated across 16 different realizations of the PI training.

network can accumulate errors (coming either from imperfect reafference or imperfect integration) for a certain number of time-steps without any noticeable effect. In the limit case where the direct model performs perfectly and reafference is exact, no resetting is ever necessary. A direct way to limit the accuracy of the direct model is to add noise to the reafferent action during training, and our hypothesis is that as this noise increases the value of the resetting gate will get closer to $0$, meaning that the resettings will be stronger and "keep less memory" of the state before resetting. This hypothesis is confirmed by Figure 16**A**. Additionally, we expect that if, at test time, we present the gating module $\mathcal{G}$ with increasingly perturbed representations, the value of the reset gate will increase until no resetting happens ($g = 1$) if the image is completely shuffled.[12] This situation is represented in Figure 16**B**.

# Reset gate neurons

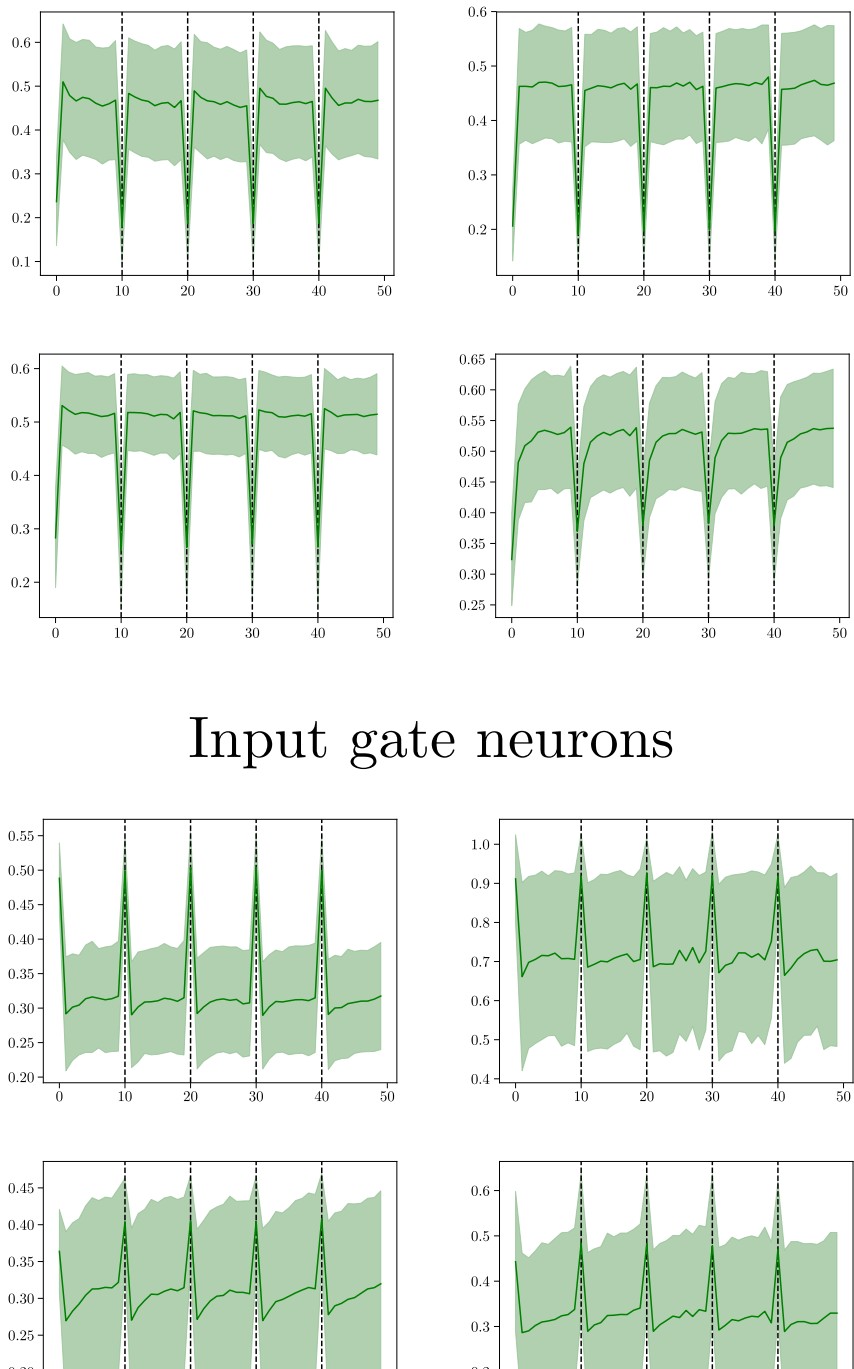

# Input gate neurons

Figure 17: Activity of four representative neurons in the "reset" and "input" gates, as a function of time, aggregated across 128 trajectories in which images are always presented at the same time-steps.

## J    INTERNAL GATES IN AN LSTM NETWORK

Averaged across a large number of trajectories, the values of the input and reset internal gates at each neuron show, to a varying extent, the behavior that was to be expected from the gate names: the reset gate neurons are inhibited when an image is presented (meaning that the previous internal state is suppressed), while the input gate neurons are activated (meaning that the current input contributes more to the internal state update). We present in Figure 17 a few representative neurons in both populations. Given the high variability that is observed in the reset and input gates, we expect that those two subnetworks contribute not only to the resetting, but also to the computation of the direct model.

## K    RESETTING IN THE CASE OF AMBIGUOUS DOUBLEDONUT

In this section, we consider the same end-to-end training procedure as in the main article, but apply it to a more complex environment comprised of 16 rooms, two of which (the left-most and right-most of the middle row) provide the agent with identical visual cues, creating an ambiguity where two different positions in the global environment correspond to the same images. This ambiguity is still such that inverse model is unambiguously defined (since there are no positions in the environment that could be reached in a single transition from both ambiguous rooms), and the forward model too as long as we choose the start position in ny non-ambiguous rooms (because otherwise, the same initial state and the same action could lead to two different new states, one for each room). As shown in Figure 18, the Resetting Path Integrator models still manage to perform reasonably well despite this ambiguity by creating new representations, distinct from the visual ones, as shown in Figure 7, and not performing resetting when the image comes from one of the ambiguous rooms, see Figure 19.

---

[12]We did not present the networks with partial shufflings of the representations at any stage in the training, only unperturbed or fully shuffled.

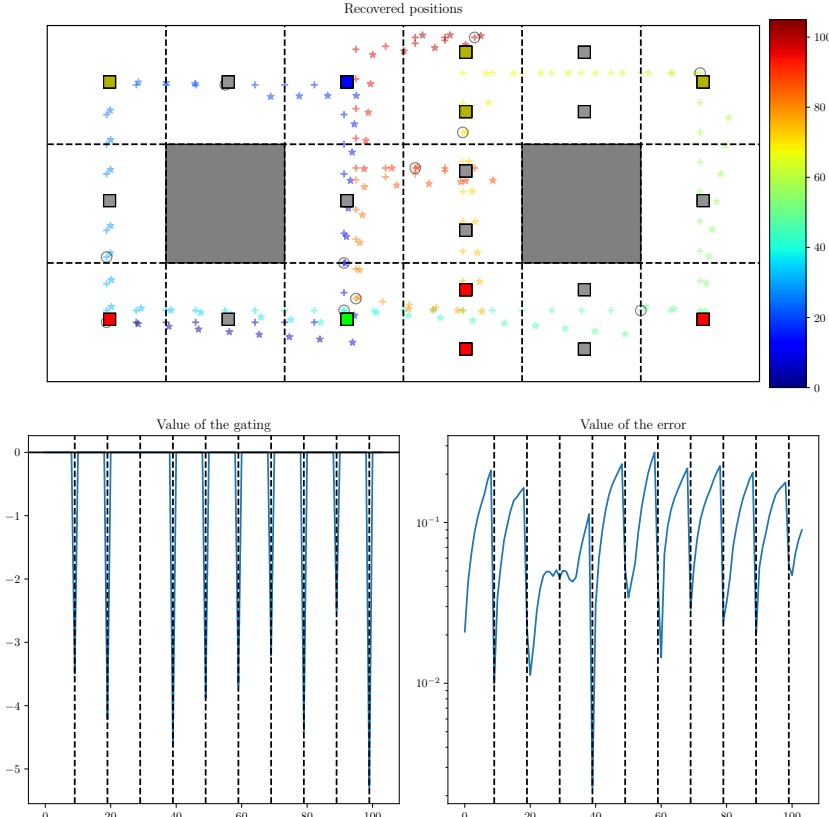

Figure 18: Example of Path Integration trajectory on the Ambiguous DoubleDonut environment. The network does not exhibit any particular drop in performance upon entering either of the ambiguous rooms, suggesting that the internal state it constructed during Path Integration lifted the ambiguity that is present in the visual cues. It still remains notable that no resetting is performed if the visual cue, even unperturbed, comes from an ambiguous room, a phenomenon further illustrated in Figure 19.

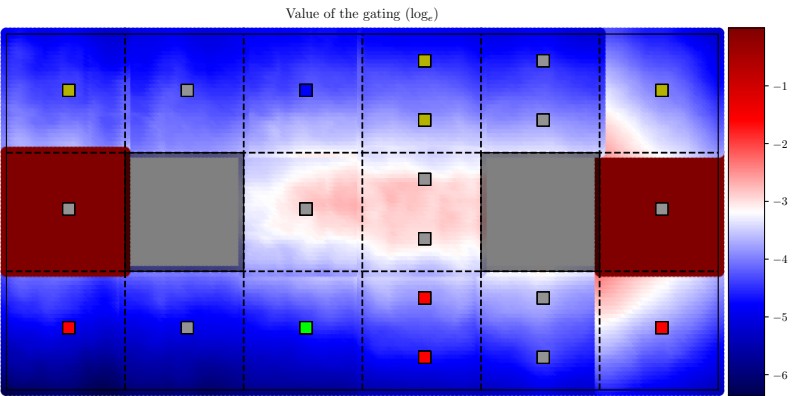

Figure 19: Value of the natural logarithm of the resetting gate $g$ as a function of position, averaged across 8 realizations of the training. As expected, resetting happens at least partially at every position in the environment, except within the two rooms that have ambiguous visual cues.

