# OpenReview forum: "Stable cognitive maps for Path Integration emerge from fusing visual and proprioceptive sensors"
_ICLR.cc/2022/Conference — ICLR 2022 Submitted_

### Official Review · Reviewer_hpxs · 2021-10-30

**Correctness:** 1
**Technical Novelty And Significance:** 2
**Empirical Novelty And Significance:** 3
**Recommendation:** 5
**Confidence:** 4

**Main Review:**

The system relies on three losses; the path integration loss and the losses for the direct and inverse models respectively. The most interesting contribution is the gating network that selects either the visual state or forward predicted state for integration. In the experiments, it is shown how the gating network helps the system to reset the state when the visual input is unique to a particular location while relying on forward prediction in case of ambiguities. This is similar to the use of so-called keyframes in SLAM that are used for similar purposes.

The PI loss assumes that the true sum of all actions, or equivalently the location at each point, is available during training, which seems to be a critical assumption in practice. Given the system’s ability to reset, errors cannot accumulate more than what is done between each reset. In a practical scenario, whether it is a biological or an artificial agent, it is hard to see how this would work. In that sense, this is different from what is assumed in SLAM.

Interestingly, the learned state depends not just on the particular location, but also on how the agent ended up there. When there are no ambiguities the gating allows the state to depend more directly on the visual input, while the opposite is true when there are ambiguities. The key component here seems to be the inverse model that is trained both to predict the action between states and the sum of actions back to the starting point, which forces the state to include this information.

The fact that the inverse model is supposed to work from the current space all the way back to the starting state, suggests that the corresponding neural network should have little opportunities to generalize beyond the starting states used during training, at least as long as there is not enough training data to avoid memorization. Since no information on how much data is used for training, one cannot really tell. However, since not only the PI loss matters, the inverse and direct models seem to introduce regularization that prevents pure memorization.

It is a bit unclear whether the proposed system is intended to model biological systems or to be used for artificial or robotic ones. Given how the conclusions are written and how noise is added to both the proprioceptive and visual signals to better emulate a real system, the biological connection seems to be most relevant. However, the simulated system is only shortly evaluated with respect to the degree of noise in Figure 6, which is left without much comment. It is also hard to tell whether the added noise really makes the simulation more realistic since no other data is used to support that.

A weakness with the proposed formulation is that both direct and inverse models are deterministic, even if a probabilistic formulation would be more suitable, where the variance could be used as an indication of confidence which could affect gating. However, a probabilistic model would most likely be harder to analyse and require more training data.

Something that strikes one as odd is why P(a) maps an action a to the same dimension as V(s) does for a state s. Is this really necessary? Is P needed at all, since it is only two-dimensional? After all, P is used in (4), but not in (7). In most cases, the perceptual space is much larger than the action space, and there is nothing in the direct model D that should require the representations to be of the same dimensionality.

A point that needs to be clarified is whether (9) is correct or whether Figure 4 is so. Currently, they do not show the same thing. By the way, there seems to be one circle missing to the left in Figure 5.



**Summary Of The Paper:**

This paper proposes a model for path integration that combines visual and proprioceptive information, i.e. a model that keeps a prediction of where an agent is located, given the past sequence of actions and observations. What is proposed is to learn paired direct (forward) and inverse models (Wolpert, 98), where the direct model predicts perceptual changes due to actions and the inverse model the actions responsible for an observed perceptual change. Through a gating mechanism information is selected from either the current location or from the path towards the location, which in effect resets path integration when unique locations are visited while overcoming ambiguities that might appear elsewhere.

**Summary Of The Review:**

The author should make it clearer what the expected target audience is, those who are interested in modelling biological systems, those interested more in artificial systems or those who search for inspiration in biological systems to build artificial ones. The paper does have its merits for each group. It might not be directly applied for e.g. robotics, but the fact that you can learn a state space that combines local and global information as suggested in this paper is worth reading about and could possibly lead to more applicable solutions in areas beyond biological modelling.

---

> ### Author Response · Authors · 2021-11-15
> **Answer to reviewer hpxs (Part 2/2)**
>
> > It is a bit unclear whether the proposed system is intended to model biological systems or to be used for artificial or robotic ones. Given how the conclusions are written and how noise is added to both the proprioceptive and visual signals to better emulate a real system, the biological connection seems to be most relevant. However, the simulated system is only shortly evaluated with respect to the degree of noise in Figure 6, which is left without much comment. It is also hard to tell whether the added noise really makes the simulation more realistic since no other data is used to support that.
>
> This concern is shared with the other reviewers, and we will address it in a revision of the introduction. As for Figure 6, we will add a short comment to make clearer its meaning: that reafference errors and integration errors have qualitatively similar effects (this is further illustrated by Appendix I, figure 16, which shows that small enough reafference errors have no effects on gating (and similarly on errors, although not shown here). This illustrates a form of robustness, which while not surprising deserves to be illustrated.
>
>
> > A weakness with the proposed formulation is that both direct and inverse models are deterministic, even if a probabilistic formulation would be more suitable, where the variance could be used as an indication of confidence which could affect gating. However, a probabilistic model would most likely be harder to analyse and require more training data.
>
>
> Indeed, this was one of our initial ideas: instead of resetting with a convex combination of proposed and observed states, we could choose between the two using G as the probability of resetting; this approach would perhaps be more suitable to analogies with cognitive theories, although neither really satisfyingly represents the underlying biological processes. Training with probabilistic resetting is possible using a Reinforce-type algorithm, but would require significant rework of the code and is not realistic within the discussion period, so it will remain a future direction  which we will mention in conclusion.
>
> > Something that strikes one as odd is why P(a) maps an action a to the same dimension as V(s) does for a state s. Is this really necessary? Is P needed at all, since it is only two-dimensional? After all, P is used in (4), but not in (7). In most cases, the perceptual space is much larger than the action space, and there is nothing in the direct model D that should require the representations to be of the same dimensionality.
>
> In theory, there is no reason that the dimensions are exactly equal, this is more of an "ad hoc" choice. In fact, allowing training of P or not does not significantly impact performance. While we did not test it, we expect that using directly the 2D action in the concatenation would result in a form of "information dilution", where only a very small fraction of the input neurons hold information about the action which could make it harder to extract said information. Finally, the fact that P is non-linear makes the "action encoding" a bit less simplistic than being directly equal to the action, which we understand could be thought of as cheating (our action input is a high-dimensional, non-linear encoding of action, not a very rich and simpe encoding).   The absence of P in (7) is a typo, which we will correct.
>
>
> > A point that needs to be clarified is whether (9) is correct or whether Figure 4 is so. Currently, they do not show the same thing. By the way, there seems to be one circle missing to the left in Figure 5.
>
> Our apologies, indeed we mixed the two, we will correct it (and P is also missing, same as in (7)). A full point seems to be missing, we will update the figure accordingly.
>
>
> > The author should make it clearer what the expected target audience is, those who are interested in modelling biological systems, those interested more in artificial systems or those who search for inspiration in biological systems to build artificial ones. The paper does have its merits for each group. It might not be directly applied for e.g. robotics, but the fact that you can learn a state space that combines local and global information as suggested in this paper is worth reading about and could possibly lead to more applicable solutions in areas beyond biological modelling.
>
>
> We agree with this observation, our introduction in particular did not properly set the expectations for the rest of the work: we are trying to propose a minimal model of Spatial Navigation, which aims at being as simple and interpretable as possible while reproducing the main properties of biological navigation systems. We do not expect to reproduce the full spectrum of position-sensitive neurons observed in biology (some recent results in this domain will be referenced), nor to improve the performance of state-of-the-art artificial navigation system.

---

> > ### Comment · Reviewer_hpxs · 2021-11-22
> > **Response to rebuttal**
> >
> > We thank the authors for addressing the questions raised and the revised manuscript. The revised introduction, in particular, more clearly places the work into context.

---

> ### Author Response · Authors · 2021-11-15
> **Answer to reviewer hpxs (Part 1/2)**
>
> > The system relies on three losses; the path integration loss and the losses for the direct and inverse models respectively. The most interesting contribution is the gating network that selects either the visual state or forward predicted state for integration. In the experiments, it is shown how the gating network helps the system to reset the state when the visual input is unique to a particular location while relying on forward prediction in case of ambiguities. This is similar to the use of so-called keyframes in SLAM that are used for similar purposes.
> The PI loss assumes that the true sum of all actions, or equivalently the location at each point, is available during training, which seems to be a critical assumption in practice. Given the system’s ability to reset, errors cannot accumulate more than what is done between each reset. In a practical scenario, whether it is a biological or an artificial agent, it is hard to see how this would work. In that sense, this is different from what is assumed in SLAM.
>
>
> We agree with you that this is a key assumption in both our approach, and the more general PI literature. In fact, it is one of the main arguments against the widespread hypothesis that PI is used as the basis for cognitive mapping and spatial navigation. One alternative which we will explore in follow-up studies is to replace the PI loss with a RL loss, for example related to finding the exit in a maze. In that case, the "metric" structure will no longer be imposed by the sum of all actions, but rather by the time to reach a goal, which is much more biologically relevant; comparing systems trained only on this RL loss and ones pretrained (or concurrently trained) for PI should shed some light on the relevance of PI for navigation.
>
> There is, however, another interpretation, which is "self-supervision": if we assume that the agent already has access to an integrator network, it can use it to generate the PI supervision signal (there will be some errors in this signal, but those could be accounted for by our "reafference errors"); in turn, this PI task would not be the end-goal, but rather a proxy task for "creating a relevant cognitive map of a new environment" (which we show is possible, using a combination of PI, direct and inverse modeling): the integrator (and the PI task self-supervision) could be seen as an example of "meta-learning" (a transferable procedure to map any new environment).
>
> > Interestingly, the learned state depends not just on the particular location, but also on how the agent ended up there. When there are no ambiguities the gating allows the state to depend more directly on the visual input, while the opposite is true when there are ambiguities. The key component here seems to be the inverse model that is trained both to predict the action between states and the sum of actions back to the starting point, which forces the state to include this information.
> The fact that the inverse model is supposed to work from the current space all the way back to the starting state, suggests that the corresponding neural network should have little opportunities to generalize beyond the starting states used during training, at least as long as there is not enough training data to avoid memorization. Since no information on how much data is used for training, one cannot really tell. However, since not only the PI loss matters, the inverse and direct models seem to introduce regularization that prevents pure memorization.
>
> This information is indeed missing: we choose starting states randomly in the environment, only avoiding the ambiguous rooms. Since the environment is continuous, there is never really any repeated starting point, but indeed enough training steps were used that the agent can generalize between training start positions. Also, it might be important to note that in the "inverse loss", the states used are always part of a single transition tuple, hence close in the environment; in the PI loss, we use the same network on states separated by an arbitrary number of steps, so the "tuning" of the network for these pairs of states is done only through the PI loss.

---

### Official Review · Reviewer_PuPV · 2021-11-04

**Correctness:** 2
**Technical Novelty And Significance:** 2
**Empirical Novelty And Significance:** 1
**Recommendation:** 5
**Confidence:** 2

**Main Review:**

#### Strengths

- The design and motivation of the proposed method is well grounded in cognitive science, which is quite interesting.
- The paper is well written, and there are sufficient details about the experiment design in the appendix.


#### Weaknesses:

- I am not exactly sure if ICLR is the right venue for this paper. Particularly, the problem that the authors study is a toy problem for navigation, it is 2D navigation with obstacles. Maybe it is common in the cognitive science community to use such environments to study the emergence of meaningful representations of the spatial structure of the environment. It seems in the AI community the focus is on complex 3D navigation problems[1]. I would recommend the authors to run the experiments on [1] so that the contribution of this work can be better evaluated.
- Secondly, the network design is rather simplistic, it is a recurrent neural network with a single gate for resetting with external signals. It is not clear if this architecture will work beyond simple 2D navigation tasks.
In the abstract the author mentioned “state-of-the-art” LSTM for 2D navigation/localization problems, I am not sure where is this coming from. For example in [1], CMP performs better than LSTM.
- In the embodied AI community, there are a lot of works on cognitive mapping (see [1][2][3]), but this work didn’t mention those, or compare with those. That’s also why I am not sure about the scope and audience of this paper. In other words, this paper might be aiming at understand biological systems, but how they can be used in AI agents or robotics is not immediately clear.

#### References

- [1] Gupta, Saurabh, et al. "Cognitive mapping and planning for visual navigation." Proceedings of the IEEE Conference on Computer Vision and Pattern Recognition. 2017.
- [2] Zhang, Jingwei, et al. "Neural slam: Learning to explore with external memory." arXiv preprint arXiv:1706.09520 (2017).
- [3] Chaplot, Devendra Singh, et al. "Learning to explore using active neural slam." arXiv preprint arXiv:2004.05155 (2020).


**Summary Of The Paper:**

The paper focuses on fusing external signals and proprioceptive signals for navigation tasks. They propose a direct-inverse model of environment dynamics to fuse image and action related signals. They propose an architecture, namely Resetting Path Integrator (RPI), that can easily and reliably be trained to keep track of its position relative to its starting point during a sequence of movements. RPI updates its internal state using the proprioceptive signal, and resets it when the image signal is present. The architecture outperforms LSTM in performance and interpretability, when benchmarked on a 2D navigation/localization problem.


**Summary Of The Review:**

This work proposes a simple yet effective way to fuse external signals and proprioceptive signals. It is benchmarked on a 2D maze-like environment and shows better performance than LSTM. I am not able to fully understand the contribution of the paper since it doesn't compare with other state-of-the-art cognitive mapping methods, and it doesn't use standard benchmarks.

---

> ### Author Response · Authors · 2021-11-15
> **Answer to reviewer PuPV**
>
> > I am not exactly sure if ICLR is the right venue for this paper. Particularly, the problem that the authors study is a toy problem for navigation, it is 2D navigation with obstacles. Maybe it is common in the cognitive science community to use such environments to study the emergence of meaningful representations of the spatial structure of the environment. It seems in the AI community the focus is on complex 3D navigation problems[1]. I would recommend the authors to run the experiments on [1] so that the contribution of this work can be better evaluated.
>
> We agree with you that 3D environments should be the end-goal here; however, these experiments require significantly more computational resources, and we expect them to be much harder to perform even from the point of view of hyperparameter optimization. The point of this first study was to ensure that our approach, using a minimal model of resetting, was suitable at least in a simple 2D, random policy setting before moving on to the follow-up studies we suggest in conclusion (transferring knowledge about the environment to accelerate learning of a spatially-structured RL task, moving to 3D and more realistic movement schemes to allow for the emergence of the wide variety of spatially tuned cells observed in biology).
>
> We insist that despite the tables of performance comparisons, our argument is not that this exact architecture should be used in real-world robotic applications; they are merely ablation studies, here to support our claim that our minimal model exhibits the desired properties of a stable cognitive map (resetting and tuning to absolute position rather than relative position with respect to the beginning of the trajectory), while off-the-shelf solutions do not.
>
> > Secondly, the network design is rather simplistic, it is a recurrent neural network with a single gate for resetting with external signals. It is not clear if this architecture will work beyond simple 2D navigation tasks. In the abstract the author mentioned “state-of-the-art” LSTM for 2D navigation/localization problems, I am not sure where is this coming from. For example in [1], CMP performs better than LSTM.
>
> We understand that state-of-the-art might not be appropriate for the reasons you mention; we will replace with "off-the-shelf" which we think better reflects the reason these networks were used. As for the simplicity of the setup, it is by design, as we want to show 1) that not all refinements of, for example, LSTM are necessary to achieve satisfying results (and in fact, better qualitative results from the point of view of representations, despite an undeniably lower expressiveness); 2) that some properties of biological systems can be replicated by minimal models through very simple "first principles". In particular, bridging the gap between artificial and biological systems would require significant simplifications, and our approach aims at going in this direction (even though we do not try to replicate all the intricacies of biological cognitive maps).
>
> > In the embodied AI community, there are a lot of works on cognitive mapping (see [1][2][3]), but this work didn’t mention those, or compare with those. That’s also why I am not sure about the scope and audience of this paper. In other words, this paper might be aiming at understand biological systems, but how they can be used in AI agents or robotics is not immediately clear.
>
>
> As mentioned by other reviewers, we failed to properly contextualize our work in a way that makes its contributions to both computational neuroscience and Machine Learning clear, and will therefore rework our introduction. We also want to emphasize that our objective is in simplification of systems for increased interpretability, which makes it far from state-of-the-art in terms of raw performance in key robotics benchmarks.

---

> > ### Comment · Reviewer_PuPV · 2021-11-24
> > **Thanks for the update**
> >
> > I would like to thank the authors for answering my questions and addressing my concerns. I think the revised manuscript positions this work better among ML and neuroscience community. In light of the revision, I am happy to revise my rating. That being said, I share with other reviewers about the lingering concerns around literature for cognitive mapping (especially in the ML community), and clarity of results of experiment details.

---

### Official Review · Reviewer_7DcK · 2021-11-06

**Correctness:** 2
**Technical Novelty And Significance:** 2
**Empirical Novelty And Significance:** 2
**Recommendation:** 3
**Confidence:** 3

**Main Review:**

In principle, a study of this type  is well suited to ICLR, and I think some of the analysis of internal dynamics in particular could be interesting. However the manuscript was difficult to follow at parts, and was missing critical details that make it impossible to evaluate. I also have reservations about the significance of problem being studied, as a full discussion of contemporary RL agents that can perform path integration is lacking. I discuss major and minor issues below. I would be happy to take a look at a revised version, as I feel some of the issues with the manuscript are clarity, but my gut feeling is the project may take more time to mature before acceptance is warranted.


Major issues
(0)	My central issue with this work is whether the RPI architecture that is the main focus of the paper is interesting because it reaches state of the art precision on a key benchmark (path integration), or whether it is interesting because it is tractable to analysis. I am not convinced that this is a state of the art network for an important task, or conversely that the results of the network analysis are interesting enough to warrant acceptance on their own. This should be made clear.

(1)	Insufficient description of past work in reinforcement learning. While there is ample discussion of the problem of path integration in nervous systems, there is very little context on approaches in artificial systems, which is the main focus of the present manuscript. Autonomous naviation is a rich field, and while some references to recent work are given in the discussion, little is given in the introduction. This makes it difficult to ascertain the novelty of the proposed architecture, and assess how the RPI fits into other contemporary models for navigation.

(2)	Missing critical details about network training and task setup. I found it difficult to evaluate many of the claims in the paper. I could not find critical details about how the network was trained using RL and associated hyperparameters. I am missing an exact definition of the action space – can both the direction and angle be arbitrarily chosen? Relatedly in Figure 5 the network is described as using “specific actions, chosen to force exploration” this process is unclear, making it difficult to evaluate the proposed claims. In Figure 2 the output of this network is unclear. Is it outputting the final action? Is it also trained to output the state? I understand that Figure 3 is demonstrating the use of forward and inverse models, but I am not exactly sure how it fits in with Figures 2 and 4.


(3)	Analysis of internal network properties. I am a bit confused about the coefficient of determination results in Table 1. I am not actually sure how these are computed. For instance with the activity-position correlation, how are these combined across the x and y dimensions? For the visual correlations, why do those vary in the LSTM cases? Wouldn’t one expect the visual networks to be similar across all the architectures? It is hard to evaluate because we are not given information on how the agents were trained, i.e. does this positional tuning just arise because these agents are heavily overtrained in a single environment? The R2 values are also very high. Even though it is clear that Figure 7 units are spatially tuned, it is not obvious that they would show an R2 of 0.99.

The link to cognitive maps is also a bit unclear here and overstated. The suggestion is that a global ‘place field’ like cell that is learned is necessary for a cognitive map (i.e. the RPI units with high R2 to position), but many other representations, include those in egocentric coordinates (ie grid cells) are also suitable as cognitive maps. As the authors note, decoding spatial position would be more convincing, as would showing that these representations could be utilized in different tasks, i.e. the classic tolman shortcut experiments.



Minor points
Pg 19 – table reference in appendix G is missing
Pg 9 – table reference is missing
I am confused about the term retina and the idea of this as a visual circuit. The information being received is highly local, making this more like a whisking system or some sort of tactile system.
Figure 7 – how were these examples chosen?


**Summary Of The Paper:**

This paper introduces several artificial models of path integration and discusses their relationship with biological systems. They train agents that receive noisy proprioceptive information and nearby visual input to minimize their path integration error when navigating in a continuous 2D environment. They compare two different architectures for path integration, based on a vanilla and modified LSTM (the resetting path integrator) that both receive proprioceptive and local visual input processed by two separate MLPs. Both architectures received intermittent visual feedback and used a gating mechanism to replace the internal state when feedback was present. They assessed what loss functions realized improved performance on the task, and found their RPI model showed improvements compared to LSTM variants. Finally, they analyzed the structure of internal representations in the network, showing networks capable of path integration showed high correlation with the absolute position in the environment, indicating that an internal representation of position was learned.

**Summary Of The Review:**

In principle, a study of this type  is well suited to ICLR, and I think some of the analysis of internal dynamics in particular could be interesting. However the manuscript was difficult to follow at parts, and was missing critical details that make it impossible to evaluate. I also have reservations about the significance of problem being studied, as a full discussion of contemporary RL agents that can perform path integration is lacking.

---

> ### Author Response · Authors · 2021-11-15
> **Answer to reviewer 7DcK (Part 4/4)**
>
> > The link to cognitive maps is also a bit unclear here and overstated. The suggestion is that a global ‘place field’ like cell that is learned is necessary for a cognitive map (i.e. the RPI units with high R2 to position), but many other representations, include those in egocentric coordinates (ie grid cells) are also suitable as cognitive maps. As the authors note, decoding spatial position would be more convincing, as would showing that these representations could be utilized in different tasks, i.e. the classic tolman shortcut experiments.
>
> We understand how our interpretation could be seen as an overstatement, in particular because of the prominence of R2 scores and their high values. In networks which have qualitatively linear activities (which in our experiments was always the case, either as a function of distance from trajectory start or absolute coordinates), they are a proxy for the regularity of these representations, which is why their exact values is relevant and computed for individual neurons;  more importantly, which of those two scores (relative or absolute) is close to 1 indicates whether the internal state represents position or relative displacement. We acknowledge in the text that non-linear coding schemes could exist with both scores low, but here we never find any so the state is always a linear function of either position or displacement.
>
>
> > Minor points Pg 19 – table reference in appendix G is missing Pg 9 – table reference is missing I am confused about the term retina and the idea of this as a visual circuit. The information being received is highly local, making this more like a whisking system or some sort of tactile system.
>
>
> We acknowledge our analogy is not the most straightforward: the way the position of the objects is mapped to the sensor state is inspired by the visual system (Difference of Gaussian fields), but the environment is seen "from above" and only in a small window across the agent, similar to a top-down perspective videogame (which we had in mind designing the environment; the screen displays the map on a small window across th e player, and actions are the 2D displacement of a joystick)
>
> > Figure 7 – how were these examples chosen?
>
> They were chosen from the first 10 neurons in the population, with differently oriented fields.

---

> > ### Comment · Reviewer_7DcK · 2021-11-22
> > **Thank you for the responses and the revised manuscript**
> >
> > Thanks to the authors for the detailed reply and new revisions to the manuscript. I think the revised introduction better positions the impact the manuscript is trying to have, as a toy model of path integration. While the authors have made some improvements (e.g. in adding details of hyperparameters improvements and training), I still think the manuscript lacks sufficient clarity in describing the results that make it difficult to evaluate, and needs to be better situated in existing literature that discusses the emergence of spatial representations and cognitive maps for navigation. In particular, as other reviewers have mentioned it is not clear that the type of spatial representation that emerges satisfies criteria as a cognitive map (e.g. flexible utility; Behrens et al. 2018 Neuron).

---

> ### Author Response · Authors · 2021-11-15
> **Answer to reviewer 7DcK (Part 3/4)**
>
> > (3) Analysis of internal network properties. I am a bit confused about the coefficient of determination results in Table 1. I am not actually sure how these are computed. For instance with the activity-position correlation, how are these combined across the x and y dimensions? For the visual correlations, why do those vary in the LSTM cases? Wouldn’t one expect the visual networks to be similar across all the architectures? It is hard to evaluate because we are not given information on how the agents were trained, i.e. does this positional tuning just arise because these agents are heavily overtrained in a single environment? The R2 values are also very high. Even though it is clear that Figure 7 units are spatially tuned, it is not obvious that they would show an R2 of 0.99.
>
> This concern is also shared with ReviewerCUYu and ReviewerAjJT. At the lowest level, this score is computed for an individual neuron as the coefficient of determination of the linear regression between the neuron state (output) and the 2D position (input); this position can be either the global position in the environment, or the relative position with respect to the start of the trajectory. These "individual neuron" scores are then averaged across all neurons in all realizations of the training. We also agree that the visual network architecture is identical, but the following layers are very different (either in architecture, or in training procedure), which explains the differences in the observed solutions after GD.
>
> ReviewerAjJT also pointed out that some details on training are missing: training is explicitly supervised, on random trajectories, using a weighted sum of PI, direct and inverse losses. The latter will be mentioned in the main text, while more details will be added in the Appendix C on network architectures. Indeed, the agent is trained in a single environment, and the subject of how to encode multiple environments in a single population is extremely exciting (notably, the capacity of such networks, ie how many cognitive maps can be learned at a given resolution). These would be very challenging studies, requiring much higher computational power than we have access to (as well as a way to procedurally generate environment). We hope that our initial results could serve as motivation to undertake this kind of study in groups who have the means to this goal.
>
> Also, Figure 7 represents units in a network trained in the ambiguous environment, which is much more challenging than SnakePath (the one of figure 5) and therefore yields lower values of R2 (which were not computed as not all networks converged). Already, switching to the unambiguous version of DoubleDonut (appendix, table 4) yielded R2 of .97 and not .99, so we expect the neurons in the ambiguous version to have even lower scores.

---

> ### Author Response · Authors · 2021-11-15
> **Answer to reviewer 7DcK (Part 2/4)**
>
> > (2) Missing critical details about network training and task setup. I found it difficult to evaluate many of the claims in the paper. I could not find critical details about how the network was trained using RL and associated hyperparameters. I am missing an exact definition of the action space – can both the direction and angle be arbitrarily chosen? Relatedly in Figure 5 the network is described as using “specific actions, chosen to force exploration” this process is unclear, making it difficult to evaluate the proposed claims. In Figure 2 the output of this network is unclear. Is it outputting the final action? Is it also trained to output the state? I understand that Figure 3 is demonstrating the use of forward and inverse models, but I am not exactly sure how it fits in with Figures 2 and 4.
>
>
> We need to draw attention here to the fact that there was no RL in this study: the network is trained in a purely supervised way: the loss function is explicitly the difference between the estimated displacement and the ground truth; no reward is used; the agent "policy" is purely random, akin to "free foraging" in rodents for example. The MDP (or, more precisely, POMDP) formalism is used only for notation convenience and to hint at future directions of research (namely, leveraging learned cognitive maps to accelerate learning of spatially-structured RL tasks). The actions are directly the 2D vector of movement: the agent can move in arbitrary direction (the "foraging policy" is simply a 2D uncorrelated Gaussian of reasonably small amplitude), so angular position is not explicitly represented. As mentioned in the response to ReviewerCUYu, the simplicity of our environment is a major limitation that we acknowledge, as most of the cell types usually associated to cognitive maps lose meaning in our setup.
>
> In figure 5, the foraging policy is replaced by another policy (at test time only), to help illustrate the behavior of the system on large displacements.
>
>
>
>
> > In Figure 2 the output of this network is unclear. Is it outputting the final action? Is it also trained to output the state? I understand that Figure 3 is demonstrating the use of forward and inverse models, but I am not exactly sure how it fits in with Figures 2 and 4.
>
> Figure 2 is a general representation of a recurrent network used for PI (the shared high-level structure between RPIs and LSTMs), and therefore aims at outputting the sum of actions since the beginning of the trajectory. Figure 4 focuses on the particular case of RPI, which also aims at outputting this sum; however, it uses a specific architecture, in which modules from the direct-inverse model presented in Figure 3 are reused: V and P to obtain the visual/action representations (not represented), D to propose the new internal state, I to deduce the total displacement from the current internal state and the "stored" initial representation. Together, Figures 3 and 4 explain the full computation diagram of the RPI module, from which the direct-inverse (fig3) and PI (fig4) losses can then be obtained.

---

> ### Author Response · Authors · 2021-11-15
> **Answer to reviewer 7DcK (Part 1/4)**
>
> > In principle, a study of this type is well suited to ICLR, and I think some of the analysis of internal dynamics in particular could be interesting. However the manuscript was difficult to follow at parts, and was missing critical details that make it impossible to evaluate. I also have reservations about the significance of problem being studied, as a full discussion of contemporary RL agents that can perform path integration is lacking. I discuss major and minor issues below. I would be happy to take a look at a revised version, as I feel some of the issues with the manuscript are clarity, but my gut feeling is the project may take more time to mature before acceptance is warranted.
>
> We thank you for your willingness to discuss, as well as for your interest in our approach. We hope the following discussion will improve your opinion of the paper.
>
>
> > Major issues (0) My central issue with this work is whether the RPI architecture that is the main focus of the paper is interesting because it reaches state of the art precision on a key benchmark (path integration), or whether it is interesting because it is tractable to analysis. I am not convinced that this is a state of the art network for an important task, or conversely that the results of the network analysis are interesting enough to warrant acceptance on their own. This should be made clear.
>
> We agree that we failed to adequately convey why our architecture (and training procedure) are relevant. The task of Path Integration that we describe here, in particular in our simplistic setting, is far from a key benchmark, and does not have direct implications in either robotics or RL; however, spatial navigation is an active field of research in neuroscience (as illustrated notably by ReviewerCUYu), where PI is often "hand-wavily" presented as a necessary ingredient to the elaboration of cognitive maps, without actually showing this is true (or, alternatively, starting from the location-specific cells which we hope to see emerge from first principles). We also acknowledge that (especially since we only find very simple global place fields) the present results are not revolutionary from the biology point of view, but they show a first step towards understanding these systems with minimal assumptions and motivate the pursuit of similar goals in more realistic (but obviously more challenging) 3D settings.
>
> > (1) Insufficient description of past work in reinforcement learning. While there is ample discussion of the problem of path integration in nervous systems, there is very little context on approaches in artificial systems, which is the main focus of the present manuscript. Autonomous naviation is a rich field, and while some references to recent work are given in the discussion, little is given in the introduction. This makes it difficult to ascertain the novelty of the proposed architecture, and assess how the RPI fits into other contemporary models for navigation.
>
> We understand this concern, and will incorporate this into our rework of the introduction.

---

### Official Review · Reviewer_CUYu · 2021-11-06

**Correctness:** 2
**Technical Novelty And Significance:** 2
**Empirical Novelty And Significance:** 1
**Recommendation:** 3
**Confidence:** 4

**Main Review:**

Pro:
- Apart from some confusion with the terminology in the introduction, the paper is clear to read and there is an abundance of analysis performed.
- The idea of learning state that encode the absolute position using path integration is interesting and potentially of great impact

Cons:
- In the introduction the authors assert that previous memory trajectories are part of proprioception, as currently written this is very inaccurate. Also, vision inputs are not allocentric, but egocentric. Broadly speaking allocentric representations are found in the hippocampal–entorhinal regions after visual stimuli have been converted in the retrosplenial cortex where it’s possible to find egocentric representation. For these reasons, I found the first paragraph poorly written and I would encourage rewriting, maybe after having read some appropriate papers that have not been cited (e.g. Spiers and Barry, 2015. Moser et al., 2018)
- Again, in the last part of the “Path integration task” section, allocentric information is used in the wrong way.
- When presenting the environment the authors describe the retinal inputs using a parallel with a monkey; however, most of the previously cited literature is on mice/rodents, this is awkward. To mitigate this I would cite Akam and Kullmann before.
- When explaining the network architecture the authors claim: "In practice, naive attempts at performing PI using the concatenation of proprioceptive and visual signals yield very unsatisfying cognitive maps which depend on position along the trajectory, but not on absolute position." It’s unclear if this is a statement about what has been tried by the authors or what is present in the literature. In case of the latter, I would recommend the author to look at the citations I'm pointing out below.  Also it is not clear what they mean by naive.
- Why H_0 is initialized with a concatenation of v_(s_0) with itself? is it one for the state and one for the cell of the LSTM? This should be clarified or motivated.
- What happens if the images are provided at irregular intervals?
- The authors used R^2 to determine whether the absolute position of the agent could be predicted from the artificial neurons. If this is applied directly on the embeddings then the authors should adjust it to account for the number of predictors in the model, which in this case  is high-dimensional. It’s not clear from the text  whether this was done or not so it should be clarified. Another option would be to use a linear decoder.
- I couldn’t find anywhere the details of the hyperparameters used in the work
- As a general comment there have been several recent papers that used unsupervised signal to learn allocentric representations (e.g. Buria et al, 2020;  Bicanski and Burgess, 2018; Whittington et al, 2018; Whittington et al, 2020) none of these work is actually cited

Minor
Authors claim that a follow up direction could be to extend this work in 3D environments. However using inverse dynamic models in 3D could be more tricky due to an increased state aliasing.


**Summary Of The Paper:**

The authors propose a network that combines propioception information with visual inputs to estimate displacement along a trajectory. The  new proposed network uses an inverse dynamic model and it is trained to do path integration. Such architecture and training regime result in a network that learns to reset and presents an internal state which correlates with absolute position in the environment.

**Summary Of The Review:**

The authors claim to have obtained stable cognitive maps in a network trained to do path integration. However this claim is not supported by the results, as they show only a correlation between learnt embeddings and location in the environment. A cognitive map is a spatial knowledge about the environment, that could be used to guide behavior in a flexible manner. From the neuroscience literature when know that this knowledge is based not just on absolute position (place cells), but also on
- an allocentric sense of direction (head direction cells)
- boundary position (boundary vector cells)
- and cells that support coding of metric distances as the animal moves through the world (grid cells).

This if we only report the most well known. So there is a lot more than just absolute location, but none of these features are present in the paper. For this reason I don’t think the authors can claim to have learnt cognitive maps.

---

> ### Author Response · Authors · 2021-11-15
> **Answer to reviewer CUYu (Part 2/2)**
>
> > I couldn’t find anywhere the details of the hyperparameters used in the work
>
> We will add values of the parameters in Appendix C about network architectures.
>
> > As a general comment there have been several recent papers that used unsupervised signal to learn allocentric representations (e.g. Buria et al, 2020; Bicanski and Burgess, 2018; Whittington et al, 2018; Whittington et al, 2020) none of these work is actually cited
>
> We thank you for the suggested references, which will be included in the new version of our paper, see below for more details.
>
> > Minor Authors claim that a follow up direction could be to extend this work in 3D environments. However using inverse dynamic models in 3D could be more tricky due to an increased state aliasing.
>
> We agree that this will be significantly more challenging, both theoretically and numerically, which is why we performed these preliminary experiments in 2D. However, inverse model can still be defined on individual transitions by suitably restricting angular and linear velocities.
>
> > A cognitive map is a spatial knowledge about the environment, that could be used to guide behavior in a flexible manner
>
> We argue that a linear representation of absolute position is a significant step in that direction.
>
> > From the neuroscience literature when know that this knowledge is based not just on absolute position (place cells), but also on
>     - an allocentric sense of direction (head direction cells)
>     - boundary position (boundary vector cells)
>     - and cells that support coding of metric distances as the animal moves through the world (grid cells).
> This if we only report the most well known. So there is a lot more than just absolute location, but none of these features are present in the paper. For this reason I don’t think the authors can claim to have learnt cognitive maps.
>
>
> We are aware of the existence of these types of cells, which explains why we mention them both in introduction and conclusion, the latter of which explicitly acknowledges that the representations we observe lack the diversity observed in biological agents. We emphasize that our minimal model was not intended to reproduce known  biological features of the mammalian brain e.g. type of cells, which is much more related to the recent works that you point to (Bicanski & Burgess 2018; Uria et al 2020). We propose a realistic explanation based on the limitations of our environment models (e.g. head-direction cells would not make sense in a model where the agent cannot rotate...). However, as you mention above, "a cognitive map is a spatial knowledge about the environment, that could be used to guide behavior in a flexible manner", the cells you mention are only one particular scheme of representing such knowledge, but not the only one (at least in our environment).
>
>
> Finally, we do not understand the score of 1 "The main claims of the paper are incorrect or not at all supported by theory or empirical results" that you attributed to the paper's correctness. Your comments have been mostly geared towards presentation details, and we feel that such a strong statement could benefit from more comments on what exactly is incorrect or unsubstantiated. Similarly, you estimate that "The contributions are neither significant nor novel"; the large amount of papers you provided suggests that this line of research is relevant, and significant differences in experimental setup exist with each of the mentioned publications (although their very successful approaches indeed need to be acknowledged in our paper): the work of Uria et al. 2020 uses a much more involved processing pipeline and does not incorporate Path Integration explicitly; similarly, the Bicanski and Burgess paper uses a very detailed and biologically relevant model, but also a much more complicated training procedure than our minimal approach; the last two papers consider only discrete spaces, both for actions and inputs, while our approach is continuous). Despite its much more restricted scope, our work is therefore not a subset of the ones you mentioned.

---

> > ### Comment · Reviewer_CUYu · 2021-11-22
> > **updated manuscript**
> >
> > I'd like to thank the author for updating the manuscripts. I did read it again and I appreciate the extra clarity on the terms in the introduction.
> > However as I pointed out in my previous review I don't think the central claim of the paper 'learning cognitive maps' holds. What the authors are doing is using the path integration objective to learn unambiguous representation of an environment. However, in my opinion, this is just a first step towards learning a cognitive map, and most likely the trivial one, as shown by other works I posted in my previous review. If you want to claim to have learnt a cognitive map then you shouldn't just provide a disambiguation experiment, but you should test your architecture on tasks that require the flexible use of this knowledge, e.g. shortcut.
> >
> > Having said that, in light of the updated version I will partially revise my scoring.

---

> > > ### Author Response · Authors · 2021-11-22
> > > **Reply to Reviewer CUYu's comments**
> > >
> > > We thank you for taking the time to read our modified manuscript, and updating your review accordingly.
> > >
> > > We emphasize that, in our work, cognitive maps are representations of the spatial structure of an environment, which are needed to support navigation but are not a navigating system per se. One goal of our experiments was to show that representations of the metric (by opposition to topological) structure of the environment could emerge in the internal state of a Path Integrator. The presence of a metric map (notably, not impacted by the presence of obstacles, as shown by the experiments we have done in the SnakePath environment) agrees with Tolman's classical "sunburst maze" experiments, in which navigation in a corridor allows for the establishment of a representation of space that contains angular information even on paths that were not seen during training. However, we did not build (nor do we claim to have) a complete navigation system, with heuristics such as shortcut exploitation, that uses this representation for trajectory planning. Information is undeniably there, in a way that could be exploited by Neural Networks, but building the associated planner (eg through RL of spatially structured tasks) remains an exciting future direction.

---

> ### Author Response · Authors · 2021-11-15
> **Answer to reviewer CUYu (Part 1/2)**
>
> > In the introduction the authors assert that previous memory trajectories are part of proprioception, as currently written this is very inaccurate. Also, vision inputs are not allocentric, but egocentric. Broadly speaking allocentric representations are found in the hippocampal–entorhinal regions after visual stimuli have been converted in the retrosplenial cortex where it’s possible to find egocentric representation. For these reasons, I found the first paragraph poorly written and I would encourage rewriting, maybe after having read some appropriate papers that have not been cited (e.g. Spiers and Barry, 2015. Moser et al., 2018). Again, in the last part of the “Path integration task” section, allocentric information is used in the wrong way.
>
> As other reviewers have also expressed concerns about the clarity of the introduction, we are rewriting it and are introducing references to the suggested literature.
>
>
> > When explaining the network architecture the authors claim: "In practice, naive attempts at performing PI using the concatenation of proprioceptive and visual signals yield very unsatisfying cognitive maps which depend on position along the trajectory, but not on absolute position." It’s unclear if this is a statement about what has been tried by the authors or what is present in the literature. In case of the latter, I would recommend the author to look at the citations I'm pointing out below. Also it is not clear what they mean by naive.
>
> This concern is also shared with ReviewerAjJT, we have therefore rewritten this sentence to adequately reflect that this is indeed a statement about results presented after the introduction and detailing what we mean by "naive".
>
>
> > Why H0 is initialized with a concatenation of $v(s_0)$ with itself? is it one for the state and one for the cell of the LSTM? This should be clarified or motivated.
>
>  We will add a footnote clarifying that the second copy is used as a reference for later (which, as we find, is a necessary condition to the emergence of resetting), while the first one is the internal state of the network (in our RPI model, there is a single internal state, which is different from LSTM) which will then be updated by the dynamics described in the following equations.
>
>
> > What happens if the images are provided at irregular intervals?
>
> During training (and, in fact during all tests except the one of Figure 6), the intervals are irregular, since "each image is presented with a probability 0.2" and not "images are presented every 5 steps". The idea of changing this probability along trajectory could be addressed, but we do not see what additional information it could bring compared to the study in Appendix I where the frequency is changed.
>
> > The authors used R2 to determine whether the absolute position of the agent could be predicted from the artificial neurons. If this is applied directly on the embeddings then the authors should adjust it to account for the number of predictors in the model, which in this case is high-dimensional. It’s not clear from the text whether this was done or not so it should be clarified. Another option would be to use a linear decoder.
>
> As we mention in the text, the methodology is that for each neuron, we compute R2 which quantifies the quality of a linear decoding of neuron state from the position (the opposite approach to decoding position from single neuron state, which would not make sense since decoding linearly a 2D quantity from a 1D vector is not possible). Linear decoding of position from the population level would be possible, but would not give any information about individual neuron regularity, which is somehow represented in this "individual" R2 score. We expect the population R2 to be significantly higher than the mean of the individual ones, since as you mention there are a large number of neurons so even if individual activities were not linearly correlated with position the population state could be linearly decoded with great accuracy. Finally, for every condition we tried, we observe that either the "absolute" or "relative" R2 is close to 1, and which one is a relevant indicator to whether the system is sensitive to relative or absolute position; this is the main point of these R2 scores.

---

### Official Review · Reviewer_AjJT · 2021-11-07

**Correctness:** 2
**Technical Novelty And Significance:** 2
**Empirical Novelty And Significance:** 3
**Recommendation:** 5
**Confidence:** 3

**Main Review:**

### Main Review


Studying spatial navigation within biological systems is interesting, especially from the computational learning perspective taken in this paper. To make an impactful contribution in this space requires a strong grasp of the cognitive science and the statistical machine learning literature, which the authors clearly have demonstrated. Critically, contributions in this space need to show that they can both fit within an existing conceptual framework of biological research, and provide a computational / statistical benefit.

The ideas in this paper are conceptually well-grounded. However, in the manuscript's current form, the empirical support is too weak to sufficiently support its claims. For this reason and several others I expand on below, I believe this paper is not ready for publication. However, I am open to changing my score depending on the responses, and the degree to which my concerns can be addressed.

### Detailed review


**The architecture** described in Figure 2 fuses information from two sensory streams. A visual stream is encoded using a convolutional neural network then concatenated with information coming from a proprioceptive stream that has been encoded with a fully-connected network. Architectures that encode then concatenate multiple sensory streams has been extensively studied. See [1] for a recurrent architecture that fuses visual and auditory streams, see [2] for an instance combining two visual streams, see [3] for an architecture that combines visual, tactile, and proprioceptive streams, and [4] for a comprehensive survey of other examples.

The paper claims that
> naive attempts at performing PI using the concatenation of proprioceptive and visual signals yield very unsatisfying cognitive maps which depend on position along the trajectory, but not on absolute position.


Since the proposed architecture (RPI) concatenates proprioceptive and visual signals, I think the paper is taking issue with the way these signals could be encoded, or rather how they would not be encoded in a naive approach. Can the authors please clarify the point they’re trying to make? Based on prior work mentioned above, any approach that concatenates encoded signals would be natural.  What are the qualities of a cognitive map that would be absent from this approach?

Finally, the joint encoding is used as input to a recurrent neural network then trained to predict total displacement.

**Training the network** is accomplished with two loss terms that enforce a reconstruction of encodings to respect the temporal structure of a Markovian observation process.  A couple things about this confused me:


1. The observation process is formalized as a Markov decision process. However, the paper does not mention anything about a reward signal, which is a critical feature of this formalism. Is the reward signal relevant to this work?
2. The paper describes some shortcomings of seperately training their encoding networks. However, it was never clear how the separate loss terms (4) and (5) were combined so the full network could be trained end-to-end.

One of the main points of novelty in this work is **a resetting mechanism**—introduced to control the internal recurrent state and allow the system to use predicted encodings from the transition model or the direct encoding of the next state. Resetting is implemented with a gating network that computes a convex combination of the prediction and the next state's encoding. The paper implicitly claims that this mechanism is beneficial in reducing the accumulation of prediction errors, presumably whenever the prediction is more accurate than the direct encoding. A few questions:


- Why would the model's prediction of the next state encoding be more accurate than the encoding that is grounded in direct experience? Please correct me here if I’m misunderstanding the significance.
- How is the gating network trained to impose beneficial resets?

Another point made in this section gave me pause.
> We therefore expect the internal state of the network to strongly depend on the current value of the position, but not on the trajectory used to get there


How can you guarantee that the internal recurrent state doesn’t depend on the history of states? Isn’t the purpose of recurrence to account for the past?


### Empirical Results


The main hypothesis claims the proposed RPI architecture can integrate long paths with less error than architectures without a resetting mechanism. The experiments consider a two-dimensional domain and compare path integration errors in two data regimes, with over trajectories of 5 and 100 time steps, respectively. Results are tabulated with uncertainty quantified over eight trials.

In addition, the experiments compare performance of each system trained on different losses. In one case, an inverse model is used, and in the other case the inverse model is absent.

Generally speaking, the experiments seemed to ask most of the right questions. There appears to be some positive support for the utility of resets within Table 1. However, it’s not clear if the results are entirely positive because most of the confidence intervals overlap. In addition, I struggle to understand the significance of the second experiment (comparison of losses). I also have a few issues involving the evaluation, baselines, and significance of some results which I describe below.

In general, more information is needed to understand the baselines. The significance of this experiment is less clear to me

There are two baselines that are necessary to establish the utility of the proposed architecture.


1. RIP with [\mathcal{G}=0](tex://\mathcal{G}=0). The recurrent state always uses the direct encoding.
2. RIP with [\mathcal{G}=1](tex://\mathcal{G}=1). The recurrent state always uses the prediction.

Comparing to the first baseline establishes that the system can benefit from prediction. In the second case—which I believe is similar to the Vanilla LSTM—the comparison implies something about the benefit of resets (i.e. direct experience). To fully support the main hypothesis, the data would need to show the learned RPI system achieves a lower path integration error than these two extremes.  Does the Vanilla LSTM reflect the [\mathcal{G}=1](tex://\mathcal{G}=1) case?

In both cases a new observation is provided every five steps. This is different from the standard Markov environment described in the formalism; where the system gets an observation at each step. I have a feeling like this property of the observation process is critical to the utility of resets. Can the authors comment on this and describe how they believe the system would perform as this hyperparameter varies?

I have several issues with Table 1. and how it’s presented. It is unclear what the errors represent and what sources of randomness are reflected between experimental trials. The experiments only used eight trials, and it is apparent from the way confidence intervals overlap that more should have been used. I suggest this experiment is rerun with thirty or more trials to improve significance.

Additional claims regarding the significance of the [R^2](tex://R^2) comparisions were less clear. What conclusion should be drawn from these results?

In Section 4.2 an experiment was performed to in situations with ambiguous observation information. The line of questioning here was not explicit or clear from the writing. Can the authors elaborate on the purpose and significance of these results (Figures 6 and 7.) What does a positive result look like in Figure 7?

### Additional Comments and Questions

- The mathematical notation was effective, even though it was ambiguous at times.
- Clarity could be improved if variables such as [\mathbf{\Delta r}_t](tex://\mathbf{\Delta r}_t) were mathematically defined.
- How is the average over trajectories computed? Are errors first summed then averaged?
- Consider adding a requirements.txt file to make it easier for others to install code dependencies and run your code.
- There are many references to the Appendix; it would help if more information was brought into the main text or the additional material was excluded.
- The related work gave sufficient coverage of the cognitive sciences, but perhaps less to the ML audience members of ICLR.


> We insist that our observations, while processed by a convolutional network, are not standard images in a ”human-readable” format, but rather a spatially organized array of sensors.


Can you expand on the difference between your “spatially organized” observations and the observations coming from a standard pixel grid?

### References


[[1] Crossmodal Attentive Skill Learner](https://arxiv.org/pdf/1711.10314.pdf)

[[2] Multimodal Deep Learning for Robust RGB-D Object Recognition](https://arxiv.org/pdf/1507.06821.pdf)

[[3] Making Sense of Vision and Touch: Learning Multimodal Representations for Contact-Rich Tasks](https://arxiv.org/abs/1907.13098)

[[4] Recent Advances and Trends in Multimodal Deep Learning: A Review](https://arxiv.org/pdf/2105.11087.pdf)


**Summary Of The Paper:**

This paper considers the problem of path integration for the purpose of determining an agent’s position in space. The authors propose a recurrent neural network architecture that fuses information from two sensory streams and integrates the representation to predict total displacement from a starting point. Their main innovation is a gating network that “resets” the hidden state. The paper claims this helps to reduce the accumulation of errors compared with other non-resetting systems. To validate this claim, they compare prediction accuracy in a two-dimensional spatial domain. The paper also makes the following additional claims


1. They claim this architecture is a good candidate for a cognitive map.
2. The proposed architecture consistently shows better performance than baseline LSTM architectures on the same problem.
3. The proposed architecture’s internal dynamics are more interpretible than a baseline LSTM.
4. The proposed architecture produces higher quality representations.

**Summary Of The Review:**

I am currently **leaning to reject the paper**, because in its current form, the empirical support is too weak to sufficiently support its claims.
1. I believe the experiments need to include an additional baseline
2.  The writing should have more details about the domain and methodology.
3. Some line of empirical questioning seems disconnected from the main claims described in the motivation.
4. I cannot be sure the proposed architectural components make a statistical difference, because the reported confidence intervals overlap.

---

> ### Author Response · Authors · 2021-11-15
> **Answer to reviewer AjJT (Part 1/4)**
>
> > The architecture described in Figure 2 fuses information from two sensory streams. A visual stream is encoded using a convolutional neural network then concatenated with information coming from a proprioceptive stream that has been encoded with a fully-connected network. Architectures that encode then concatenate multiple sensory streams has been extensively studied. See [1] for a recurrent architecture that fuses visual and auditory streams, see [2] for an instance combining two visual streams, see [3] for an architecture that combines visual, tactile, and proprioceptive streams, and [4] for a comprehensive survey of other examples.
>
>
> We thank you for these references, which give more context to multimodal learning. It should be noted that those approaches are all tailored towards particular modality combinations, and in much the same way we propose an architecture (and training procedure) for a new pair of modalities (vision + movement).
>
>
> > The paper claims that naive attempts at performing PI using the concatenation of proprioceptive and visual signals yield very unsatisfying cognitive maps which depend on position along the trajectory, but not on absolute position.
>
>
> We agree that this statement lacks clarity, as has been noted by other reviewers too, and we have modified it to better reflect what was intended: "A natural approach to multimodal PI consists in simply concatenating non-recurrent encodings of action and visual inputs, and feeding the resulting joint representation to a Recurrent Neural Network trained on the PI loss (1). However, as reported in the following, these initial attempts yielded unsatisfying solutions that 1) failed to perform "resetting" (see Section 3) when an image was available, and 2) had internal states in the RNN that were correlated only with displacement from the start of the trajectory, and not with the absolute position in the environment. This second property is particularly important for a Cognitive Map, i.e. for the neural representations of an environment's spatial structure: the state of the neural population should encode the current absolute position (and not only the position within the trajectory).
>
> > Since the proposed architecture (RPI) concatenates proprioceptive and visual signals, I think the paper is taking issue with the way these signals could be encoded, or rather how they would not be encoded in a naive approach. Can the authors please clarify the point they’re trying to make? Based on prior work mentioned above, any approach that concatenates encoded signals would be natural. What are the qualities of a cognitive map that would be absent from this approach?
>
> We hope the aforementioned reformulations make these two points clearer. In particular, we take no issue with existing solutions from a theoretical point of view, but instead empirically observe that qualitatively and quantitatively better results can be obtained with specialized architectures.
>
>
> > Finally, the joint encoding is used as input to a recurrent neural network then trained to predict total displacement.
> Training the network is accomplished with two loss terms that enforce a reconstruction of encodings to respect the temporal structure of a Markovian observation process. A couple things about this confused me:
> -The observation process is formalized as a Markov decision process. However, the paper does not mention anything about a reward signal, which is a critical feature of this formalism. Is the reward signal relevant to this work?
>
> This is a very relevant question for future directions. The experiments presented here focus on a kind of "free foraging" task, in which the agent explores its environment without any explicit incentive to do so (as is observed in experiments). One theory to explain this behavior is that the agent uses these interactions to construct a cognitive map of the environment, which could then be later used for few-shot learning of new tasks (which could involve rewards, for example navigating to the position of a drop of fruit juice). However, the question of how the presence of a reward (and its prediction in the direct model) influences representations remains to be addressed in a follow-up work.

---

> ### Author Response · Authors · 2021-11-15
> **Answer to reviewer AjJT (Part 2/4)**
>
> > The paper describes some shortcomings of seperately training their encoding networks. However, it was never clear how the separate loss terms (4) and (5) were combined so the full network could be trained end-to-end.
>
>
>  This is an oversight on our part, we involuntarily cut the equation defining the total loss used for training which is simply a weighted sum of the direct, inverse and PI losses. We will make this more visible in the new version.
>
>
> > One of the main points of novelty in this work is a resetting mechanism—introduced to control the internal recurrent state and allow the system to use predicted encodings from the transition model or the direct encoding of the next state. Resetting is implemented with a gating network that computes a convex combination of the prediction and the next state's encoding. The paper implicitly claims that this mechanism is beneficial in reducing the accumulation of prediction errors, presumably whenever the prediction is more accurate than the direct encoding. A few questions:
> Why would the model's prediction of the next state encoding be more accurate than the encoding that is grounded in direct experience? Please correct me here if I’m misunderstanding the significance. How is the gating network trained to impose beneficial resets?
>
>
> In a situation where the observation from the environment is unambiguous, we agree that direct observations should always be more accurate. However, in our experiments, we consider several sources of errors on this observation process (no image, or several positions being associated with the same image). In those cases, the visual observation holds very little information about position, hence the result of the direct model will almost surely be more informative.
>
>
>
>
> > Another point made in this section gave me pause.
> We therefore expect the internal state of the network to strongly depend on the current value of the position, but not on the trajectory used to get there. How can you guarantee that the internal recurrent state doesn’t depend on the history of states? Isn’t the purpose of recurrence to account for the past?
>
>
> This is a very important point, and actually the main difference between our approach and direct concatenation. Indeed, recurrence allows for propagation of information across time-steps (as observed in the case of the ambiguous environment, this is the case even with our RPI model). The question is what information flows between time-steps through the internal state. In the case of standard concatenation (vanilla LSTM), the internal state represents directly the integral of actions, so the information that flows is the integral up to that point. In the case of RPI, because of the gating and the different losses, the internal state represents the absolute position in the environment, and the integral itself is not represented; to extract it (which, in the end, is the objective for PI), the inverse model is leveraged, reconstructing the distance between current and initial representation. While there is no \textit{a priori advantage to this "two-steps" way of reconstructing the integral for PI alone, it is precisely what allows for the emergence of a cognitive map (which we argue would be transferable towards other spatial navigation tasks).
>
>
> > There are two baselines that are necessary to establish the utility of the proposed architecture.
> Comparing to the first baseline establishes that the system can benefit from prediction. In the second case—which I believe is similar to the Vanilla LSTM—the comparison implies something about the benefit of resets (i.e. direct experience). To fully support the main hypothesis, the data would need to show the learned RPI system achieves a lower path integration error than these two extremes. Does the Vanilla LSTM reflect the G=1 case?
>
> We understand the significance of these comparisons, which are in fact subsets of already existing experiments:
>
> - if G is always equal to 0, because 4 out of 5 observations are purely random, the optimal solution for our network would be to always output $0$ in these cases, which appears qualitatively unsatisfying
>
> - if G is always equal to 1, the network simply never uses resetting, and our network reduces to a non-gated RNN (which is a priori significantly worse than LSTM, a gated RNN which has the necessary connections to implement resetting). In practice, RPIs trained without the direct-inverse losses never exhibit resetting and indeed have G=1 at all time steps. As shown in Table 1, these systems indeed achieve noticeably worse errors on long time-scales (which should be expected, as without resetting errors will accumulate indefinitely). They are also phenomenologically close to Vanilla LSTMs, which do not perform resetting.

---

> > ### Comment · Reviewer_AjJT · 2021-11-29
> > **Reply**
> >
> > > We understand the significance of these comparisons, which are in fact subsets of already existing experiments:
> >
> > The outcome from the G=0 case is not captured in the current experiments. I think this is an important baseline, because it establishes to what degree the learning system benefits from its prediction.
> >
> > > We therefore expect the internal state of the network to strongly depend on the current value of the position, but not on the trajectory used to get there.
> >
> > Sorry, but the response didn't get me any closer to understanding this point. How can you be suer the internal state is not a function of history?
> >
> > >  the visual observation holds very little information about position, hence the result of the direct model will almost surely be more informative.
> >
> > Thank you for clarifying. The hidden state can help dealias the observations.

---

> ### Author Response · Authors · 2021-11-15
> **Answer to reviewer AjJT (Part 3/4)**
>
> > In both cases a new observation is provided every five steps. This is different from the standard Markov environment described in the formalism; where the system gets an observation at each step. I have a feeling like this property of the observation process is critical to the utility of resets. Can the authors comment on this and describe how they believe the system would perform as this hyperparameter varies?
>
> Indeed, the frequency at which observations are available is crucial to understanding resetting, which is why it was studied in Appendix I: while qualitative behavior is the same, the level at which PI errors stabilize increases as his frequency decreases (since errors accumulate on a longer time-scale), and resettings when an image is finally available are stronger.  We will make it clearer in the text that this is a Partially Observable MDP.
>
> > I have several issues with Table 1. and how it’s presented. It is unclear what the errors represent and what sources of randomness are reflected between experimental trials. The experiments only used eight trials, and it is apparent from the way confidence intervals overlap that more should have been used. I suggest this experiment is rerun with thirty or more trials to improve significance.
>
> We will modify the table description to make it clearer that between trials, we change the network initial parameters and the trajectories that are sampled. Unfortunately, due to computational budget constraints and the limited time for discussion, it will be difficult for us to run 20 more realizations of all training conditions. We will try to add as many as possible in Table 1, but still argue that the overlap between the distributions observed is small enough that a trend can still be seen even with our limited sample sizes.
>
>
> > Additional claims regarding the significance of the R2 comparisions were less clear. What conclusion should be drawn from these results?
>
> The R2 score is directly related to the optimal linear decoding of each neuron's internal state from position (more details in response to Reviewer CuYu), and can be used as a proxy to estimate the regularity of the representations (in the case where, from direct observation, it appears that indeed the network adopts a linear representation scheme as is the case in our experiments, either in absolute or relative position). As we briefly mention in the text, non-linear decoding could be considered (if the question was purely on positional information content), but qualitatively it should be expected that a linear function of position will (generally) be more transferable to arbitrary spatial navigation tasks. More importantly, since for every condition either the absolute or relative R2 is close to 1, we know that the neurons encode linearly a positional feature, and the relevant question becomes "which one?".
>
> > In Section 4.2 an experiment was performed to in situations with ambiguous observation information. The line of questioning here was not explicit or clear from the writing. Can the authors elaborate on the purpose and significance of these results (Figures 6 and 7.) What does a positive result look like in Figure 7?
>
>
> We will modify the text of this section to make the question more explicit: we want to show that our model is able to replicate a known property of RNNs for Partially Observable MDPs, which is to disambiguate states that share identical observations (the images) while being different for the "hidden" part of the environment (the true absolute position). This is in fact very similar to what was done by giving the observation only at certain steps (the same observation, corresponding to no image, could be given at different random positions), but in a more intuitive and behaviorally relevant situation (for example how humans could differentitae two identical rooms by remembering the path they took to enter them).
> Figure 7 actually shows both what an inconclusive result (top row) and a conclusive result (bottom row) would look like (although top line is the visual representation layer of a positive outcome, not the recurrent state of a negative outcome): if integration of previous movement did not help lift the ambiguity, the internal representation in the two opposing rooms would be identical, which is not the case with our training procedure.

---

> ### Author Response · Authors · 2021-11-15
> **Answer to reviewer AjJT (Part 4/4)**
>
>
> > Clarity could be improved if variables such as were mathematically defined.
>
> We will add this definition in an equation form.
>
> > How is the average over trajectories computed? Are errors first summed then averaged?
>
> Yes, in practice the loss is computed for each trajectory then averaged, which as far as we know is standard practice.
>
> > Consider adding a requirements.txt file to make it easier for others to install code dependencies and run your code.
>
> We will upload it.
>
> > There are many references to the Appendix; it would help if more information was brought into the main text or the additional material was excluded.
>
> Unfortunately, we are constrained by ICLR paper length and cannot move appendices to the main text without first removing some information, which we think would make the paper less readable. While these appendices are not necessary to understand the main messages of the paper, they are still useful to consolidate its claims, so we do not think it would be good to remove them entirely either.
>
> > The related work gave sufficient coverage of the cognitive sciences, but perhaps less to the ML audience members of ICLR.
>
> Most reviewers agree on that point, which we will address in the reworked introduction.
>
> > We insist that our observations, while processed by a convolutional network, are not standard images in a ”human-readable” format, but rather a spatially organized array of sensors. Can you expand on the difference between your “spatially organized” observations and the observations coming from a standard pixel grid?
>
> We will change this sentence, as it is somehow misleading: these observations can be represented as an image (they are the state of a 3x64x64 array of neurons), the difference is that this image is not the one that is depicted in the figures (it represents a preprocessed version of that image, through Difference of Gaussians visual receptive fields).

---

### Decision · Program_Chairs · 2022-01-20

**Decision:**

Reject

**Comment:**

The paper considers the problem of path integration in cognitive maps, where combining proprioception with visual inputs is required to estimate the displacement.  The paper proposes a small mechanism (a resetting path integrator) that extends a conventional LSTM for this purpose.  The resulting networks demonstrate better performance and interpretability than a conventional LSTM on tested problems.

The reviewers raised many issues with the paper.  One concern was whether the problem was to model biological, artificial, or robot problems (reviewer hpxs, PuPV), which the authors successfully addressed by stating that it is a minimal model. Many other minor concerns were also addressed. However, significant concerns remained.  One is the emphasis on the cognitive map (reviewer CUYu, hpxs) for which path-integration is a small part.  Another major concern is the significance of the results, with reference to the baselines and $R^2$ (AjJt, CUYu).  A third is on the generalizability of the method beyond single small examples (AjJT, hpxs, CUYu).

All reviewers indicate reject due to concerns that the paper is not ready for publication.  The paper is therefore rejected.